



# Control of 3D tectonic inheritance on fold-and-thrust belts: insights from 3D numerical models and application to the Helvetic nappe system

Richard Spitz[1], Arthur Bauville[2], Jean-Luc Epard[1], Boris J.P. Kaus[3], Anton A. Popov[3], and Stefan M. Schmalholz[1]

[1]Institute of Earth Sciences, University of Lausanne, 1015 Lausanne, Switzerland
[2]Department of Mathematical Science and Advanced Technology, Japan Agency for Marin-Earth Science and Technology, 3173-25, Showa-machi, Kanazawa-ku Yokohama, Japan
[3]Institute of Geosciences, Johannes Gutenberg University Mainz, Mainz 55128, Germany

**Correspondence:** Richard Spitz (richard.spitz@unil.ch)

## Abstract

Fold-and-thrust belts and associated tectonic nappes are common in orogenic regions. They exhibit a wide variety of geometries and often a considerable along-strike variation. However, the mechanics of fold-and-thrust belt formation and the control of the initial geological configuration on this formation are still incompletely understood. Here, we apply three-dimensional (3D)

thermo-mechanical numerical simulations of the shortening of the upper crustal region of a passive margin to investigate the control of 3D laterally variable inherited structures on the fold-and-thrust belt evolution and associated nappe formation. We consider tectonic inheritance by applying an initial model configuration with horst and graben structures having laterally variable geometry and with sedimentary layers having different mechanical strength. We use a visco-plastic rheology with temperature dependent flow laws and a Drucker-Prager yield criterion. The models show the folding, detachment and horizontal

displacement of sedimentary units, which resemble structures of fold and thrust nappes. The models further show the stacking of nappes. The detachment of nappe-like structures is controlled by the initial basement and sedimentary layer geometry. Significant horizontal transport is facilitated by weak sedimentary units below these nappes. The initial half-graben geometry has a strong impact on the basement and sediment deformation. Generally, deeper half-grabens generate thicker nappes and stronger deformation of the neighboring horst while shallower half-grabens generate thinner nappes and less deformation in the

horst. Horizontally continuous strong sediment layers, which are not restricted to inital graben structures, cause detachment folding and not overthrusting. The amplitude of the detachment folds is controlled by the underlying graben geometry. A mechanically weaker basement favors the formation of fold nappes while stronger basement favors thrust sheets. The applied model configuration is motivated by the application of the 3D model to the Helvetic nappe system of the French-Swiss Alps. Our model is able to reproduce several first-order structural features of this nappe system, namely (i) closure of a half-graben

and associated formation of the Morcles and Doldenhorn nappes, (ii) the overthrusting of a nappe resembling the Wildhorn and



Glarus nappes and (iii) the formation of a nappe pile resembling the Helvetic nappes resting above the Infrahelvetic complex. Furthermore, the finite strain pattern, temperature distribution and timing of the 3D model is in broad agreement with data from the Helvetic nappe system. Our model, hence, provides a first-order 3D reconstruction of the tectonic evolution of the Helvetic nappe system based on thermo-mechanical deformation processes.

## 1 Introduction

Fold-and-thrust belts are common in nature and typically associated with orogenic belts, such as the Himalayas or the European Alps (e.g. Price and McClay, 1981; Lacombe and Bellahsen, 2016). The structural interpretation of fold-and-thrust belts is based on the interaction between the crystalline basement and the overlying sedimentary cover. Two end-member deformation styles are commonly distinguished: thin-skinned deformation, without significant basement involvement, and thick-skinned deformation, with significant basement involvement (Rodgers, 1949; Pfiffner, 2006). Due to their importance for the fundamental understanding of mountain building processes and for natural resources exploration, the formation of fold-and-thrust belts has been studied since several decades with field and modelling studies ( e.g. Davis et al., 1983; Dahlen, 1984; Dahlen and Suppe, 1988; Beutner, 1977; Price and McClay, 1981; Gillcrist et al., 1987; Butler, 1989; Ramsay, 1989; Buchanan and Buchanan, 1995: Dunn et al., 1995; Mitra, 1997; Lacombe and Mouthereau, 2002; Wissing and Pfiffner, 2003; Simpson, 2011; Yamato et al., 2011; Ruh et al., 2012; Fernandez and Kaus, 2014; Bellahsen et al., 2012; Bauville and Schmalholz, 2015; Lacombe and Bellahsen, 2016; Bauville and Schmalholz, 2017). However, the mechanical deformation processes controlling fold-and-thrust belt evolution are still incompletely understood. One challenge for understanding fold-and-thrust belt evolution is that the formation, spacing, orientation and time-sequence of thrusts, shear zones and folds are controlled by two different factors: First, the mechanical deformation behavior of rocks, which can be dominated either by brittle-frictional sliding or by ductile creep and can further be strongly affected by various mechanical softening mechanisms, such as frictional strain softening, reduction of effective friction by fluid overpressure, grain size reduction with damage or thermal softening. Second, the geometrical configuration, such as half-graben structures or orientation of sedimentary layers, and variations in rock strength, for example between basement and cover or within the cover by alternation of strong, such as carbonates, and weak, such as shales, sediments. To illustrate these two controlling factors in a simple way, let us consider the deformation of a linear viscous material under homogeneous pure shear. Adding a circular inclusion with a smaller viscosity to the viscous material will not generate a shear zone inside the linear viscous material for this deformation configuration. The only possibility to generate a shear zone in the viscous material is to add a softening mechanism, such as thermal softening (Jaquet et al., 2015; Kiss et al., 2019) or grain-size reduction with damage (Bercovici and Ricard, 2003; Austin et al., 2008). In contrast, if the linear viscous material is sheared over a non-planar interface, resembling a half-graben, then a shear zone can develop inside the linear viscous material even without any softening mechanism (Bauville and Schmalholz, 2017). Therefore, a main challenge for understanding the thermo-mechanical evolution of fold-and-thrust belts is to determine whether the major thrusts and shear zones have been controlled by a particular rheological softening mechanism or by pre-existing geometrical and mechanical





heterogeneities, referred to here as tectonic inheritance. In nature, there is most likely a continuous transition between these two controlling factors.

Many studies employing analogue and numerical models have been performed with a focus on the impact of different rheological models. Studies investigated the impact of brittle, brittle-ductile, visco-plastic and visco-elasto-plastic rheological models on fold-and-thrust belt evolution, and also studied, for example, the impact of fluid pressure and associated reduction of effective friction (e.g. King Hubbert and Rubey, 1959; Stockmal, 1983; Merle, 1989; Simpson, 2011; Ruh et al., 2014; Poulet et al., 2014; Bauville and Schmalholz, 2015; Granado and Ruh, 2019). Other numerical studies focused more on the impact of

tectonic inheritance, in the form of mechanical heterogeneities, on fold-and-thrust belt evolution (Wissing and Pfiffner, 2003; Bauville and Schmalholz, 2017). The majority of numerical modeling studies uses two dimensional (2D) models, which were often able to produce results that are to first oder comparable with geological reconstructions and cross-sections. However, it is well known that the style of fold-and-thrust belts can vary considerably along-strike the belt (e.g. Hamilton, 1988; Mitra and Fisher, 1992; Mitra, 1997; Mouthereau et al., 2002; Fitz Diaz et al., 2011; Nemčok et al., 2013). Inherited, laterally varying

pre-existing structures are important, for example, in the Alps where pre-Alpine laterally varying passive margin structure presumably exert a strong control on the deformation style (Pfiffner, 1993; Pfiffner et al., 2011). A recent study of the Iberian passive margin by Lymer et al. (2019) highlights the complex 3D architecture of such margins. Their results imply that fault systems disappear laterally or link together in lateral direction along the margin, consequently creating discontinuities and geometrical asymmetries. Therefore, it is important to consider the 3D inherited heterogeneities of passive margins when

studying fold-and-thrust belts that resulted from the deformation of passive margins, as is the case for the Helvetic fault-and-thrust belt (Pfiffner et al., 2011).

    Here, we apply a 3D thermo-mechanical numerical model to investigate the fundamental impacts of mechanical heterogeneities, representing graben structures and sedimentary layering, on the deformation style during fold-and-thrust belt formation. A particular aim is to apply our model results to the Helvetic nappe system in the Swiss-French Alps (see next section).

We employ an initial model configuration that mimicks a simplified upper crustal region of a passive margin and is composed of a basement and several sedimentary units. The passive margin contains a half-graben system that varies along the lateral direction. Moreover, we apply laboratory derived temperature dependent dislocation creep flow laws for all our model units and consider a brittle-frictional Drucker-Prager yield stress. We use a typical velocity boundary condition to simulate the large scale deformation conditions during tectonic wedge formation (Simpson, 2011; Ruh et al., 2014). In order to keep our model

relatively simple, we concentrate on the thermo-mechanical processes on the macroscale and the impact on kilometer-scale structures. Hence, we do not consider microscale processes such as grain size reduction involving secondary mineral phases and damage. Furthermore, our model does not include hydro-chemical coupling. Hence, we do not model processes such as fluid release and decarbonatization (Poulet et al., 2014).

    The aims of our study are to: (1) understand the impact of lateral changes in half-graben geometry on the deformation style,

(2) investigate the importance of the spatial distribution of a competent layer and half-graben geometry during fold-and-thrust belt formation and (3) discuss the application of our numerical models to the formation of the Helvetic nappe system.



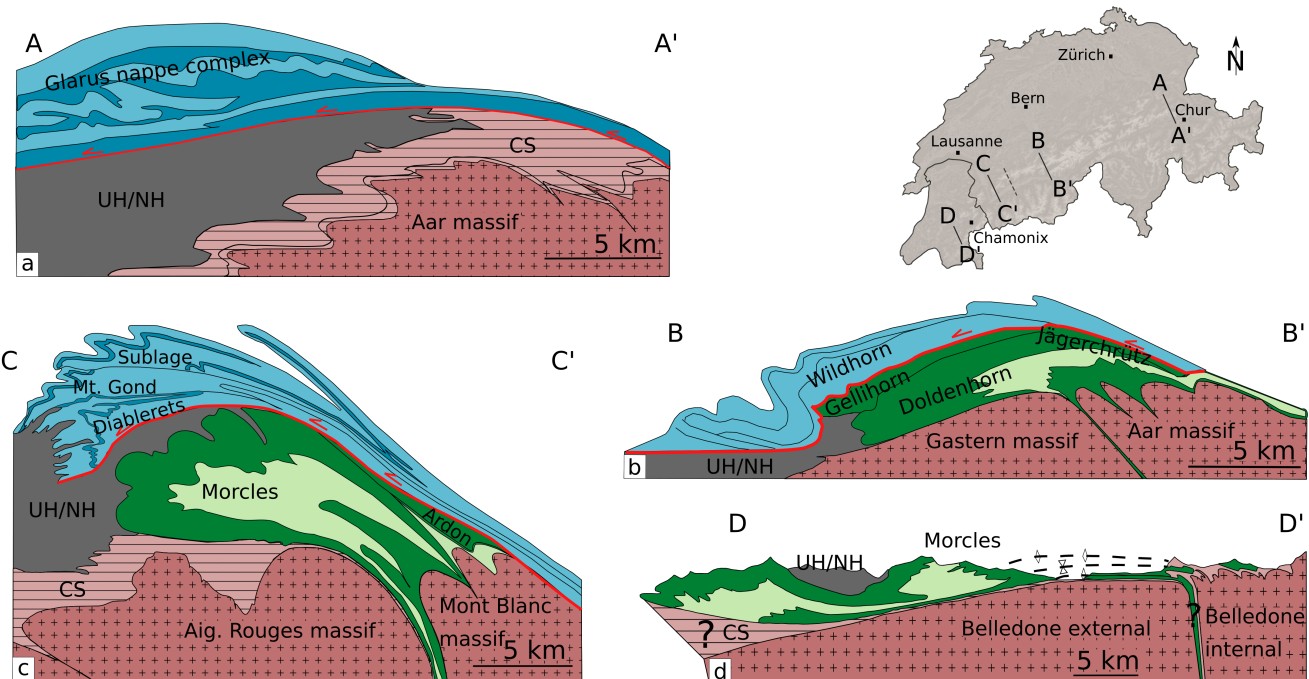

**Figure 1.** Simplified geological cross-sections along strike of the Helvetic nappe system. a) Glarus nappe complex (modified after Pfiffner, 2015). b) Doldenhorn nappe (modified after Kirschner et al., 1999). c) Morcles nappe (modified after Escher et al., 1993). d) Morcles nappe at the Belledonne massif (modified after Epard, 1990). UH/NH = Ultrahelvetics/North Helvetics. CS = Cover sediments.

## 2 Overview of the Helvetic Nappe system

The Helvetic nappe system is a fold-and-thrust belt complex that was formed during the Alpine orogeny when the European passive margin collided with the Adriatic margin (e.g. Trümpy et al., 1980; Pfiffner, 2015). The system consists of a pile of
tectonics nappes, which mainly comprise Mesozoic and Cenozoic sediments derived from the former European continental margin (Masson et al., 1980; Ramsay, 1981; Pfiffner, 1993; Escher et al., 1993; Steck, 1999; Pfiffner et al., 2011). Commonly, the nappe system is subdivided in the structurally upper Helvetic nappes, considered mainly as allochtonous thrust nappes, and the structurally lower Infrahelvetic complex, which can involve par-authochtonous fold nappes (Pfiffner, 1993; Pfiffner et al., 2011) (Figure 1).

The onset of Alpine burial of the proto-nappe system is constrained by the last deposited sediments with ages of ca. 28 to 34 Ma (Kirschner et al., 1995; Nibourel et al., 2018). Peak metamorphic conditions in the Helvetic nappe system occurred between ca. 25 and 17 Ma, indicating the end of the main phase of nappe stacking (Kirschner et al., 1995; Nibourel et al., 2018). The main phase of nappe formation and stacking occurred presumably during a period of ca. 10 to 15 Ma (Masson et al., 1980; Milnes and Pfiffner, 1980; Burkhard, 1988; (Kirschner et al., 1995; Nibourel et al., 2018). Uplift and exhumation
of the Helvetic nappe system occurred between ca. 20 Ma and today (Kirschner et al., 1995; Nibourel et al., 2018). We focus





here on the main phase of nappe formation and stacking and do not consider the subsequent uplift and exhumation of the Helvetic nappe system.

The Helvetic nappe system exhibits a wide range of nappe geometries, including two commonly considered end-member nappe styles, namely fold nappes and thrust nappes, or thrust sheets (Termier, 1906; Epard and Escher, 1996). Fold nappes

are recumbent folds with fold amplitudes of several kilometers and with a stratigraphic inversion in a prominent overturned limb. Thrust nappes are coherent allochtohonous rock sheets that are displaced along a basal shear zone and lack a prominent overturned limb.

We consider here four simplified geological sections across the Helvetic nappe system in the Swiss-French Alps and focus on several prominent nappes within these sections (Figure 1). The first order tectonic features of these four cross sections and

the associated nappes will be compared with our 3D modelling results.

The first section includes the Glarus nappe of the Eastern Swiss Helvetic nappes (Figure 1a). Geological reconstructions suggest a displacement of approximately 50 km from its original location along a thin basal thrust zone that is composed of Mesozoic sediments (e.g. Pfiffner, 2015). Studies suggest that the Glarus basal thrust originates inside Carboniferous strata allowing for the transport of the Glarus nappe consisting of Permian Verrucano units at its base (e.g. Schmid, 1975; Pfiffner,

1993; Pfiffner, 2015). Observations on the thrust zone suggest earlier viscous dominated deformation followed by dominantly brittle deformation (Herwegh et al., 2008). A number of studies investigated the complex deformation behavior of the thrust zone and suggest the involvement of pressurized fluids that resulted in hydro-fracture networks and the reduction of the friction at the base (e.g. Burkhard et al., 1992; Badertscher and Burkhard, 2000; Badertscher et al., 2002; Herwegh et al., 2008; Hürzeler and Abart, 2008). Recently, Poulet et al., 2014 suggest a superposition of viscous and brittle deformation mechanisms due to

ductile shear heating resulting in decarbonatization and the release of overpressurized fluids causing brittle fracturing.

The second section includes the Doldenhorn nappe, belonging to the Infrahelvetic complex, which has been overthrusted by the Wildhorn nappe, belonging to the Helvetic nappes (Figure 1b). The Doldenhorn nappe consists of Mesozoic and Cenozoic parautochthonous sediments that have been squeezed and sheared out of a pre-Alpine half-graben, referred to here as North Helvetic basin (Figure 2). The Gellihorn and Jägerchrütz nappes are minor nappes and their sediments are considered as

deposits on a basement high, likely a horst, which seperated the half-graben including the Doldenhorn sediments from the more distal marginal basin, referred to here as Helvetic basin, on which the Wildhorn sediments have been deposited (Masson et al., 1980) (Figure 2). The Doldenhorn nappe roots in the Aar basement massif. Studies indicate metamorphic peak temperatures in the Doldenhorn nappe of up to $380°C$ (Herwegh and Pfiffner, 2005; Ebert et al., 2007a). These temperatures allowed for ductile deformation and folding of the Doldenhorn nappe during nappe formation. Colder temperatures around ca. 250 $°C$ in

the structurally higher Wildhorn nappe were likely responsible for a deformation style resembling more a thrust nappe.

The third section includes the Morcles fold nappe (Figure 1c) belonging to the Infrahelvetic complex. It is overlain by a major thrust nappe, which is termed in this region the Wildhorn super-nappe (Steck, 1999). The term super-nappe is used, because the nappe can be subdivided, from bottom to top, into the Diablerets, Mont-Gond and Sublage nappes. Similarly to the Doldenhorn nappe, the Morcles nappe is considered as the result of the closure of the North Helvetic basin and the

subsequent extrusion of sediments during compressional Alpine tectonics (e.g. Ramsay, 1981; Pfiffner, 1993). This North





Helvetic basin comprised kilometer thick sequences of shale-rich units with competent carbonate units in between. Different to the Doldenhorn nappe, the Morcles nappe exhibits less pronounced shearing at its base, a more prominent overturned limb and stronger internal isoclinal folding. The strongly thinned and overturned limb is in contact with the crystalline Aigulles-Rouges massif below and its autochthoneous sediments. Between the Morcles and Wildhorn nappe is a minor sedimentary

nappe, the Ardon nappe, which is considered as originating from the horst region, from a similar paleogeographic position as the Jägerchrütz nappe (Figure 2). Estimates of metamorphic peak temperatures range between $250°C$ and $380°C$ and therefore support a dominantly ductile deformation regime (Leloup et al., 2005; Boutoux et al., 2016). Furthermore, the deformation of the Morcles nappe is constrained by finite strain measurements. The data highlights a pattern of increasing strain from the front of the nappe towards the root zone and also from the top to the bottom. Strain ellipses show X/Y ratios > 400 at the contact

between the overturned limb and the basement-cover (Ramsay and Huber, 1987; Casey and Dietrich, 1997). Microstructural observations of the basal mylonitic shear zone in the overturned limb of the fold nappe indicate ductile creep in the calcite-rich lithologies (Austin et al., 2008). The Wildhorn super-nappe, as a whole, resembles more a thrust sheet but exhibits significant internal deformation. For example, the Diableret nappe is separated from the Mont Gond nappe by an isoclinal fold indicating significant ductile deformation inside the super-nappe.

Due to the topographic Rawil depression, there is no continuous outcrop from the Doldenhorn towards the Morcle nappe. However, geological reconstructions suggest, that the Doldenhorn and Morcles nappes originate from the same, lateral continuous North Helvetic basin and that the Wildhorn nappe and super-nappe result from the same laterally continuous Helvetic basin of the Mesozoic passive European margin (Epard, 1990). The North Helvetic basin is considered absent in the eastern region of the Helvetic nappe system, which explains the absence of significant nappes in the Infrahelvetic complex below the

Glarus thrust. The Doldenhorn nappe can be considered as an intermediate nappe type between a thrust nappe, represented by the Glarus thrust, and a fold nappe, represented by the Morcles nappe.

The fourth section includes also the Morcles nappe and is located in the French Alps (Epard, 1990) (Figure 1d). In this section, no Helvetic nappes are outcropping. In contrast to the Morcles fold nappe in the third section, the Morcles nappe is here not a fold nappe but rather a thrust nappe (cp. Figure 1c and 1d, respectively), because there is no prominent overturned

limb. The basement massif is there termed Belledone massif, but the Morcles nappe in the French region is considered as a geological continuity from the Morcles nappe in the Swiss region (Epard, 1990) (Figure 2).

Some studies suggested mechanical explanations for the variation in nappe style within the Helvetic nappe system. The formation of fold nappes in the Southwestern part of Switzerland and the lack of such fold nappes in the Northeastern part is explained by lateral variations of the mechanical stratigraphy, that is the alteration of mechanically strong, such as carbonates,

and weak, such as shales, sedimentray units (Pfiffner, 1993; Pfiffner et al., 2011). Different thickness ratios, $n$, of weak to strong sedimentary units cause a different mechanical response during shortening. Low ratios $n < 0.5$ favor imbricate thrusting and harmonic folding while higher ratios of $n$ favor fold nappes and detachment folding (Pfiffner, 1993). This impact of thickness ratios on deformation style, which was derived by field observations, is also supported by 2D numerical simulations (Jaquet et al., 2014). Moreover, Von Tscharner et al. (2016) quantified, with 3D numerical models of viscous deformation, the impact

of laterally varying half-graben depth on the folding of sedimentary layers in the half-graben. Their 3D model results also







**Figure 2.** Simplified paleogeographic map of the lower cretaceous showing the assumed distribution of the basement massifs and sedimentary units forming the discussed tectonic nappes of the Helvetic nappe system (modified after Epard, 1990 and Pfiffner, 2015). The cross section on the top left represents a section through the SW region across the Helvetic nappe system (modified after Jaquet et al., 2018).

confirm that laterally varying sediment thickness has a strong impact on fold amplification and nappe formation. However, their models do not generate thrust nappes and also not the stacking of nappes, as observed in the Helvetic nappe system. A comparison of observed finite strain gradients across the Morcles fold nappe with finite strain gradients resulting from a theoretical thermo-mechanical shear zone model utilizing calcite flow laws suggests that the Morcles fold nappe was generated

by heterogeneous shearing during Alpine shortening (Bauville and Schmalholz, 2013), as was already suggested by kinematic models (Ramsay et al., 1983; Dietrich and Casey, 1989; Casey and Dietrich, 1997).





# 3 Methods

## 3.1 Numerical method

We apply the concept of continuum mechanics to describe the deformation of rocks with a system of partial differential equations (e.g. Mase, 1970). To solve the resulting system of equations numerically, we apply the 3D thermo-mechanical parallel code LaMEM (Kaus et al., 2016; https://bitbucket.org/bkaus/lamem) for our simulations. The equations describing the conservation of mass, linear momentum and energy are:

$$\frac{\partial v_i}{\partial x_i} = 0 \tag{1}$$

$$-\frac{\partial P}{\partial x_i} + \frac{\partial \tau_{ij}}{\partial x_j} = \rho g_i \tag{2}$$

$$\rho C_p \frac{\partial T}{\partial t} = \frac{\partial}{\partial x_i}\left(\lambda \frac{\partial T}{\partial x_i}\right) + H_R + H_S \tag{3}$$

where $x_i$ (i = 1,2,3) refers to Cartesian coordinates in the three spatial directions (i=1 indicates x-direction, i=2 y-direction and i=3 z-direction), $v_i$ are the components of the velocity vector, $P$ is pressure (negative mean stress), $\tau_{ij} = \sigma_{ij} + P\delta_{ij}$ are components of the deviatoric Cauchy stress tensor (with $\delta_{ij}$ being the Kronecker delta), $\rho$ is density, $g_i = [0\ 0\ g]$ the gravity acceleration vector with $g$ being the gravitational acceleration, $C_p$ is the specific heat, $T$ the temperature and $\lambda$ the thermal

conductivity. The source term $H_R$ refers to the radiogenic heat production and $H_S = \tau_{ij}\dot{\varepsilon}_{ij}$ for shear heating. The components of the deviatoric strain rate tensor $\dot{\varepsilon}_{ij}$ are defined by the visco-plastic constitutive equations:

$$\dot{\varepsilon}_{ij} = \dot{\varepsilon}_{ij}^{vs} + \dot{\varepsilon}_{ij}^{pl} = \frac{\tau_{ij}}{2\eta_{eff}} + \dot{\varepsilon}_{II}^{pl}\frac{\tau_{ij}}{\tau_{II}} \tag{4}$$

where $\dot{\varepsilon}_{ij}^{vs}$ is the viscous strain rate tensor, $\dot{\varepsilon}_{ij}^{pl}$ is the plastic strain rate tensor, $\eta_{eff}$ is the effective viscosity, $\tau_{ij}$ are components of the deviatoric stress tensor and $\tau_{II} = (\frac{1}{2}\tau_{ij}\tau_{ij})^{\frac{1}{2}}$ is the square root of the second invariant of the deviatoric stress tensor.

The temperature dependent viscosity $\eta_{eff}$ for the considered dislocation creep is:

$$\eta_{eff} = \frac{1}{2}(B_n)^{-\frac{1}{n}}(\dot{\varepsilon}_{II})^{\frac{1}{n}-1}exp\left(\frac{E_n}{nRT}\right) \tag{5}$$

where $n$ is the stress exponent, $\dot{\varepsilon}_{II} = (\frac{1}{2}\dot{\varepsilon}_{ij}\dot{\varepsilon}_{ij})^{\frac{1}{2}}$ the square root of the second invariant of the strain rate tensor, $B_n$ the creep constant, $E_n$ the activation energy and $R$ the universal gas constant. The components of the plastic strain rate tensor $\dot{\varepsilon}_{ij}^{pl}$ are determined by enforcing the Drucker-Prager yield criterion given by:

$$\tau_{II} \leq \tau_Y = \sin(\phi)P + \cos(\phi)C \tag{6}$$

Here, $\tau_Y$ is the yield stress, $\phi$ the friction angle and $C$ the cohesion.

The system of equations is discretized with a staggered grid finite difference approach and solved with LaMEM. Material properties are advected employing a marker-and-cell method. In order to maintain computational stability for large time steps we employ a stabilized free surface boundary condition. For detailed information on the LaMEM code see Kaus et al. (2016)

and the documentation on the official website (https://bitbucket.org/bkaus/lamem).



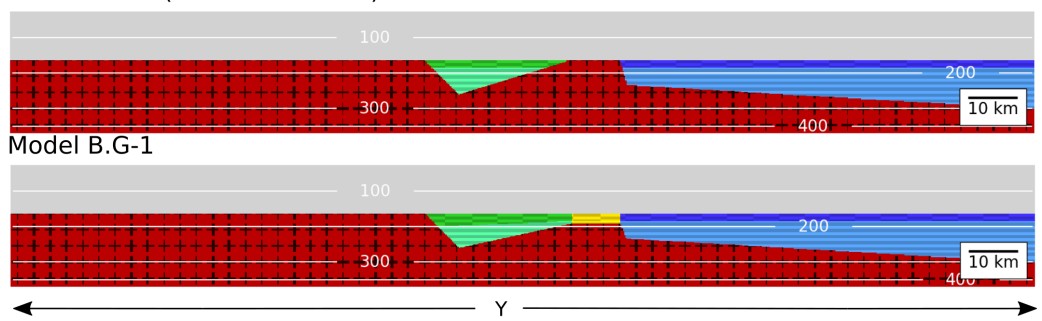

a General model configuration

b Graben variations for model A

c Side view (X = 0 km)

d Boundary conditions

e Model evaluation by X-slices:

f Graphical overlay of passive marker lines used for numerical cross-sections





**Figure 3.** a) 3D geometry of our reference model configuration using a laterally disappearing half-graben, and highlighting the distribution of our numerical phases and their material properties. Here, the color *green* refers to weak materials and *dark blue* to strong materials. The configuration is based on the paleogeographic reconstructions as indicated by the abbreviations on the model: AR = Aiguilles Rouges massif, MB = Mont Blanc massif, HB = Helvetic Basin, NH/UH = North Helvetic flysch and Ultrahelvetic units, HG = North Helvetic graben. b) Display of our three different half-graben systems used as initial setup. c) Cross-sectional view at X = 0 km for three models utilizing half-graben G-1 in combination with three different stratigraphic distributions. d) Sketch displaying our boundary conditions. e) 3 cross-sectional slices along X-direction to explain our model evaluation. f) Subfigure highlighting our passive marker overlay which is used to distinguish the different structural units HG (*green*) and HB (*blue*) from each other. Temperature isotherms are shown in [$°C$].

## 3.2 Model configuration

The model configuration (Figure 3a) has a lateral width of X = 70 km, a length of Y = 210 km and height of Z = 40 km. Our numerical resolution for most simulations is $128 \times 256 \times 128$ (X × Y × Z) grid points with a mesh refinement in Z-direction using 96 grid points between Z = 8 and 32 km and 16 grid points each for Z < 8 km and Z > 32 km. Our model consists of five units which are, from top to bottom: a sticky air, a cover, a strong layer, a weak unit and a basement unit. Each unit has distinctive mechanical properties, such as for example, a flow law, friction angle, or cohesion that corresponds to natural materials. In addition to the standard parameters of the flow laws we add a pre-factor $f$ to the creep constant $B_n$ to facilitate a brittle-ductile transition zone at a depth of $8-10$ km. The details of the mechanical properties for each unit are listed in Table 1. The initial model geometry is based on a simplified and idealized cross-section of the European crustal continental passive margin. The basement constitutes the bottom of the model domain and has a maximum height of 15 km. It involves a half-graben (HG) that is separated by a horst from a larger distal basin (HB) to the right (Figure 3b). HG represents the half-graben region in which the sediments of the Morcles and Doldenhorn nappes have been deposited, HB represents the Helvetic Basin in which the sediments of the Wildhorn nappe have been deposited and the horst between HG and HB represents the domain on which the sediments of the Ardon and Gellihorn nappes have been deposited (Fig. 2). In the reference configuration HG is becoming shallower and finally disappears towards the NE-direction (positive X-direction; Figure 3a). The half-graben, between 110 km > Y > 90 km, is subjected to different geometries in our different model configurations with respect to its lateral extend (X-direction). The total dimensions of the graben system are fixed with a maximum depth of 7 km, a length of 25 km (Y-direction) and a width of 50 km (X-direction). In contrast, the geometry of the distal basin (Y < 90 km) is constant and has no variations in the X-direction for all model configurations. Both half-graben and basin are filled with a weak unit, mimicking shale-rich sediments, that is overlain by a strong layer with a thickness of 1.5 km, mimicking strong carbonates. Additionally, we apply a 10 km thick sediment cover, mimicking the North Helvetic Flysch and Ultrahelvetic units. We assume that these cover unit have been thrust on top of the proto-Helvetic nappe system before the onset of nappe formation. Finally, we use a low density, low viscous sticky air unit ($\eta = 10^{19} Pa.s^{-1}$, $\rho = 1 kg.m^{-3}$) on top to simulate a free surface boundary condition.

In order to investigate the impact of the geometry of HG on the structural evolution of our model we use two different series of simulations. In the first one, we vary the geometry of HG in lateral direction using three different geometries (Figure 3b),





namely G-1 to G-3. Here, we also define our reference model A.G-1. It compromises HG, G-1, that linearly shallows and
       narrows out in lateral X-direction. The next configuration, G-2, is HG without lateral geometrical variations, which is bounded
       by a 90° wall to the adjacent basement. Our third configuration, G-3, is HG being a full graben that shallows out in lateral
       direction while the length in Y-direction is constant.

       In the second series (Figure 3c) we employ the same HG as in the reference configuration, G-1, but modify the spatial con-
nection between the strong layer and the underlying basement. In model B.G-1 we decrease the height of the horst in such a
       way that the strong layer and a part of the weak unit rests on the top of the horst. Additionally, we also thicken the left part
       of the strong layer which is connected to the basement in a wedge like shape (Figure 3c). In the last model, C.G-1, we extend
       the strong layer to the left of the HG (Figure 3c). For C.G-1, the layer is not connected with the basement, but underlain by an
       additional 0.5 km thick layer of weak units (Figure 3c).

To test the impact of the vertical strength distribution in the basement, we performed an additional simulations, D.G-1, with
       the same configuration as the reference model, A.G-1, but we cut-off the deviatoric stresses in the basement at 40 MPa.
       Numerically, this is done by setting the cohesion in the basement to 40 MPa and setting the friction angle to zero. Such yield
       criterion corresponds essential to a pressure-insensitive von Mises yield strength and mimics a semi-brittle deformation, or a
       low temperature plasticity.

We apply free slip boundary conditions on all sides of the model except on the top where we model a free surface boundary
       condition with the sticky air method (Figure 3d). In order to mimic the kinematic conditions during tectonic wedge formation,
       we apply a constant velocity boundary condition on the bottom face and the left XZ-face of our model (Figure 3d). We
       induce bulk shortening of the model by moving the left boundary and the bottom boundary in positive Y-direction with a
       constant velocity of $v_y = 1\,cm/yr$. This boundary condition is similar to typical sandbox analogue models of accretionary
wedge formation. The bulk shortening strain rate $\dot{\varepsilon}_{bg}$ is recalculated from the velocity boundary condition for every time
       step. The shortening of the model in Y-direction is balanced by an elongation in Z-direction, with no changes of the width in
       X-direction.

       We use a linear, vertical temperature gradient of 16.6°C/km with a fixed temperature of 20°C at the surface and 435°C at the
       bottom of the model. The heat flux through all vertical model sides is zero. Furthermore, we use passive marker lines and
patterns to improve the visualization of 2D cross-sections of our 3D model (Figure 3f). Also , we slightly change the color
       scheme to highlight the two major basement structures HG and HB, together with the sediments on top of the basement horst
       (*yellow*). The darker colors inside the HG and HB indicate the strong layer while the lighter colors correspond to the weak
       units. The layering and patterns are passivley advected with the numerical velocity field. Hence, they do not influence the
       material properties and deformation and are simply there for visualization purposes.





**Figure 4.** 3D model evolution for our six simulations, using a graphical threshold to highlight the deformation of the strong layer, weak units and the basement. Rows correspond to the configuration and columns to the total bulk shortening $\gamma_b$.



| Model unit | $\rho$ [$kg.m^{-3}$] | **Rheology** | f | A[$Pa^{-n}.s^{-1}$] | n | Q [$J.mol^{-1}$] | $\lambda$[$W.m^1 K^{-1}$] | $\phi$ [°] | C [MPa] |
|---|---|---|---|---|---|---|---|---|---|
| Cover | 2750 | Calcite[1] | 0.1 | $1.58 \times 10^{-25}$ | 4.2 | $4.45 \times 10^5$ | 2.0 | 30 | 1 |
| Strong layer | 2750 | Calcite[1] | 1.0 | $1.58 \times 10^{-25}$ | 4.2 | $4.45 \times 10^5$ | 2.5 | 30 | 1 |
| Weak units | 2750 | Mica[2] | 1.0 | $1 \times 10^{-138}$ | 18.0 | $5.10 \times 10^5$ | 2.5 | 5 | 1 |
| Basement | 2800 | Granite[3] | 1.0 | $3.16 \times 10^{-26}$ | 3.3 | $1.87 \times 10^5$ | 3.0 | 30 | 10 |

**Table 1.** Table listing the material properties of our model units for most models, where $\rho$ is the density, A is the pre-exponential factor, f is custom pre-factor, n is the power-law exponent, Q is the activation energy, $\lambda$ is the thermal conductivity, $\phi$ is the friction angle, and C is the cohesion. Some additional parameters are constant: Here the thermal expansion coefficient $\alpha = 3 \times 10^{-5}\ K^{-1}$, the heat capacity $C_p = 1050\ J.K^{-1}$ and the radiogenic heat production $Q_r = 10^{-6}\ W.m^{-3}$. We use following creep flow laws: [1]Schmid et al. (1977),[2]Hansen et al. (1983),[3]Kronenberg et al. (1990).

## 4 Results

### 4.1 3D model evolution

We first provide an overview of the general model evolution of all six performed 3D simulations. Figure 4 shows the structural evolution for three different bulk shortening for each model configuration. All models, except model C.G-1 (Figure 3c), show the formation of nappe-type structures in the strong layers and stacking of the strong layer from HB on top of the strong layer of HG. Model C.G-1 does not generate any nappe-like structure or overthrusting, but generates detachment folds.

In the first stage of our reference model A.G-1 (Figure 4a) the strong layer of HB is detached from the horst by the formation of a shear zone. Both the strong layer and the weak units below are thrusted on top of the horst culminating in a horizontal displacement of about 10 km. Our model generated a structure resembling a thrust sheet or thrust nappe. The HB experiences closure. In the region of HB, the weak sediments and the basement are thickened with ongoing bulk shortening. Initially, the basement experiences an uplift at the right boundary due to the imposed velocity discontinuity. The basement uplift increases the topography at the right boundary of the model (i.e. the backstop). The HG shows minor signs of deformation at 23%. The strong layer is partially sheared and squeezed out in front of HG. Additionally, we observe a slight depression of the strong layer in the rear of HG. In the next shortening stage, (Figure 4b) the closure and inversion of HG is in progress. The strong layer has been squeezed out of HG. The horizontal displacement of the strong layer of HG decreases in positive X-direction associated with decreasing depth of HG. Furthermore, the thrust nappe from HB has advanced on top of HG and the horst exhibits a dome-like shape. At 47 % bulk shortening (Figure 4c), the half-graben is almost completely closed resulting in the formation of a elongated cusp dipping towards the backstop. The infill of HG has been extruded and compromises a nappe that is overthrust by a nappe from HB. The front of the nappe from HB is essentially straight along the lateral, X-direction, indicating that the displacement of the nappe is essentially unaffected by the formation of the nappe below, whose front varies significantly along the X-direction. We only observe a slightly higher topography above the nappe from HG, causing a slight tilting of the nappe from HB towards the positive X-direction.





Model A.G-2 (Figure 4b) deviates in the structural evolution from model A.G-1. At 23 % bulk shortening (Figure 4b), the strong layer of HG is displaced more out of HG than for model A.G-1. The basement horst is slightly sheared towards the left, that is against the shortening direction. This shearing of the horst increases with progressive bulk shortening. At 47 % bulk

shortening (Figure 4b) the HG is nearly closed at its top, but the closure of the HG did not form a cusp, as for model A.G-1. In contrast, the sediments still in the HG become thicker with depth, because for the considered geometry of the HG it is more difficult to squeeze out all the sediments (Figure 4b). The strong layer of HG has been completely detached from the horst showing a higher slope in front of the extruded sediments. Overall, this model shows a significantly higher degree of basement involvement, with the up doming horst separating the nappe from HG from the thrust nappe from HB. Nevertheless, the front

of the nappe from HB was displaced in the same uniform manner as in the reference model, A.G-1.

Model A.G-3 displays a similar evolution as model A.G-1 (Figure 4c). The geometry of HG results in a almost straight front of the squeezed out strong layer. There is a slight curvature at the lateral boundary between the lateral end of HG and basement (Figure 4c). The final stage (Figure 4c) shows a nappe originating from HG that is almost homogeneous in lateral X-direction. However, even though the depth of the half-graben is constant up to X = 50 km there is a notable decrease of nappe height from

X = 30 to 50 km. This tilting of the nappe front towards the positive X-direction can be explained by the adjacent basement, which might affect the degree of half-graben closure due to the strength contrast between the sediment and basement units. At 47 % bulk shortening (Figure 4c) the nappe from HB did not fully thrust over the fold nappe.

Model B.G-1 (Figure 4d) exhibits a continuous strong layer from HG to HB across the horst, but has the same geometry of HG than reference model A.G-1. At 23 % bulk shortening (Figure 4d) we observe a similar deformation of the strong layer

at HG than in model A.G-1. However, a larger amount of weak sediments from HB is displaced across the horst against the shortening direction. At 35 % bulk shortening the HG is less closed that for the same bulk shortening of the reference model (Figure 4d). Inside the initially continuous strong layer, a shear zone develops that forms a nappe of sediments from the region of the HB. At 47 % bulk shortening (Figure 4d) two nappe structures have formed; one consisting of sediments from HG and a structurally higher one consisting of sediments from HB. The two nappes are also stacked. However, in this model the strong

layer has no connection anymore to the horst.

Model C.G-1 (Figure 4e), with a strong layer resting above the entire basement, shows detachment folding. The lateral variations in basement geometry are not significant enough to generate a shear zone in the strong layer, which eventually would form a nappe structure. The deformation of the basement and HG is similar to model B.G-1. The detachment fold with the largest amplitude originates from the region of HG (Figure 4e). With progressive shortening, this detachment fold is displaced

across the basement against the direction of shortening (Figure 4e).

In model D.G-1 (Figure 4f), with a stress cut-off in the basement at 40 MPa, the shortening is more homogeneously distributed in the basement, so that during the initial stages of shortening (23 %; Figure 4f) the basement at large distance from the backstop is already thickened. Consequently, the basement uplift around the backstop is significantly lower compared to all other models. Furthermore, during nappe formation some parts of the uppermost basement are also sheared-off from the basement. The nappe

forming from the strong layer of HG resembles a fold nappe.





## 4.2 2D numerical cross sections

We discuss the deformation in the 3D models by analyzing six 2D cross sections parallel to the shortening direction (Y-direction) but at six different location along the lateral X-direction (Figure 5). The six cross sections are located along the lateral direction from X = 0 km to X = 50 km with 10 km spacing.

For the reference model A.G-1 (Figure 5) at 23% shortening (Figure 5, left column), the sediments from HB have been detached from their original position and thrusted across the basement horst all along the lateral direction (everywhere in the left column of Figure 5). The displaced sediments from HB resemble a thrust nappe. Around the backstop (right side) there is significant basement uplift. The cross sections show, from top to bottom, the shallowing and disappearance of HG. The strong layer is already sheared-out of HG, whereby the horizontal displacement is larger for deeper HG. Both nappes originating from HB

and HG are deforming at the same time. The temperature around the top of the horst is ca. 300°C.

The passive marker lines in the weak sediments of HB indicate ductile flow generating passive shear folds. In the basement the passive marker symbols ('crosses') indicate an increase in shear strain from the top to the bottom of the basement due to decrease in basement viscosity associated with a temperature increase. At 47% bulk shortening (Figure 5g-l, right column) the nappe from HB is thrust completely above the nappe from HG. The HG has been almost closed with the weak sediments

residing now inside a cusp between the neighboring basement units, which dips towards the backstop. In the deeper region of HG the passive markers in the weak sediments indicate a fold nappe structure. The very frontal part of the nappe from HG resembles a thrust nappe, formed exclusively by the strong layer, whereas the main part of the nappe, including also the weak sediments, resembles a fold nappe (Figure 5g-i). Between the upper nappe from HB and the lower nappe from HG are cover sediments that have been dragged between the two nappes during overthrusting of the nappe from HB (Figure 5g-k).

The nappe from HG disappears in lateral direction with the disappearance of HG. For the cross section without HG (Figure 5l) there is only a nappe from HB resembling a thrust nappe. Despite the significant lateral variation of the depth of HG and the associated lower nappe, the horizontal displacement of the upper thrust nappe is essentially the same along the lateral direction. Consequently, nappe formation of sediments from HG does not affect the horizontal displacement of the overthrusting nappe from HB. The horst exhibits a significant internal deformation in the regions with a deep HG, indicated by the passive marker

symbols. The temperature isotherms are affected by the deformation and thickening of the model. Generally, the sediments are getting hotter during the deformation. At 23% bulk shortening the strong layer of HG has a temperature of ca. 300°C while the temperature of the strong layer of HB is less than 300°C. At 47% bulk shortening the isotherms indicate that both strong layers were heated by ca. 50°C.

Additionally to the cross sections showing the structural and thermal evolution of model A.G-1, we display the same cross

sections but indicating the magnitude of the deviatoric stress invariant, $\tau_{II}$, to quantify the state of stress (Figure 6). The largest stress is ca. 140 MPa and occurs at the brittle-ductile transition in the cover, in a depth of ca. 5 km. At 23% bulk shortening (Figure 6a-f) the top of the basement exhibits $\tau_{II}$ values between 40 MPa and 80 MPa. After 47% bulk shortening $\tau_{II}$ values in the basement are strongly decreased, down to 10 MPa to 20 MPa, due to the increase of basement temperature. Overall, there are no considerable lateral variations in $\tau_{II}$.





**Figure 5.** Graph showing lateral cross-sections for two different states of bulk shortening for the geometrical evolution of model A.G-1. Cross-sections are taken in 10 km steps from X = 0 km to X = 50 km. Additionally we display the isothermal lines in degree Celsius [°C]. Columns correspond to the bulk shortening state and rows to cross-section X-position.



**Figure 6.** Graph showing lateral cross-sections for two different states of bulk shortening for the second invariant of the deviatoric stress $\tau_{II}$ of model A.G-1. Cross-sections are taken in 10 km steps from X = 0 km to X = 50 km. Additionally we display the isothermal lines in degree Celsius. Columns correspond to the bulk shortening state and rows to cross-section X-position.





**Figure 7.** Graph showing lateral cross-sections for two different states of bulk shortening for the geometrical evolution of model A.G-2. Cross-sections are taken in 10 km steps from X = 0 km to X = 50 km. Additionally we display the isothermal lines in degree Celsius [°C]. Columns correspond to the bulk shortening state and rows to cross-section X-position.





The cross-sections for model A.G-2 indicate overall a similar structural evolution as our reference model (Figure 7) in the sense that the sediments are sheared-out of HG and HB, form nappe-like structures and are piled at 46% bulk shortening. However, the different initial geometry of HG also generates differences in the structural evolution: At 23% bulk shortening the nappe from HB is less displaced across the horst for a deeper HG. The reason is that due to the different geometry of HG, the basement horst is also significantly sheared towards the left model side and represents, hence, a less stiff obstacle for a

deep HG. Therefore, both the nappe from HB and the horst are sheared together towards the left model side. For a shallower HG, the horst represents a mechanically stiffer obstacle for the nappe from HB and, hence, the nappe is overthrust more to the left. At 46% bulk shortening (Figure 7g-l) we also observe extrusion of sediments from HG, but for the deepest regions of HG a significant amount of the sediments remains trapped between the basement at depth. Around the region where HG is deepest, the horst is sheared significantly and even sheared slightly on top of the basement that was initially to the left of

HG (Figure 7g-i). The strong layer from HG is essentially disconnected from the horst and a significant part of this strong layer resembles more a thrust nappe. The temperature evolution of model A.G-2 is similar to the one of the reference model. The results indicate that the geometry of HG has a strong impact of the structural evolution of the nappes and the basement, although the first order structural evolution is similar to the reference model.

Model A.G-3 (Figure 8) shows a similar structural evolution as the reference model. The main difference can be observed in

the cross section located at the lateral boundary of HG at X-position 50 km (Figure 8l). There, sediments of HG have been extruded laterally out of HG, which can be seen by a pocket of sediments inside the basement.

Model B.G-1 (Figure 9) shows a different nappe evolution than model A.G-1. At 23% bulk shortening (Figure 9a-f) in the region where the horst is overlain by weak sediments and a strong layer (*yellow*), the strong layer is sheared off the horst and pushed across HG by the strong layer from HB (Figure 9a). In the region where the horst is overlain only by the strong layer

(*yellow*), the strong layer is continuously sheared and dragged by the sediments from HB (Figure 9b). The degree of shearing decreases with decreasing thickness of the yellow layer (Figure 9b-d). At 47% bulk shortening, HG is also almost closed and nearly all sediments have been squeezed out (Figure 9g-f). At X = 0 km (Figure 9f) the strong layer is completely detached from the basement. The layer initially resting on the horst has been sheared above the sediments from HG and is itself overthrust by the sediments from HB. This structure resembles the vertical stacking of three nappes whose sediments were originally

horizontally next to each other. With decreasing thickness of the strong layer from the horst, its shearing across the sediments from HG is also decreasing (Figure 9i-k). Despite a horizontally continuous strong layer from HG to HB, a major shear zone developed in the strong layer and caused the generation of a nappe-like structure. The shear zone development inside the strong layer is only due to the geometrical variation of the underlying basement, because the strong layer has homogeneous material properties.

Model C.G-1, with a continuous strong layer across the entire model domain (Figure 10), shows a very different evolution compared to the reference model, because no prominent shear zones form in the strong layer, which could develop a nappe-like structure and significant overthrusting.

The strong layer initially above HG develops a detachment fold. At 23% bulk shortening (Figure 10a-f) this detachment fold already shows a great variability in amplitude in lateral X-direction of the model. During the initial stages of folding, the





**Figure 8.** Graph showing lateral cross-sections for two different states of bulk shortening for the geometrical evolution of model A.G-3. Cross-sections are taken in 10 km steps from X = 0 km to X = 50 km. Additionally we display the isothermal lines in degree Celsius [°C]. Columns correspond to the bulk shortening state and rows to cross-section X-position.



**Figure 9.** Graph showing lateral cross-sections for two different states of bulk shortening for the geometrical evolution of model B.G-1. Cross-sections are taken in 10 km steps from X = 0 km to X = 50 km. Additionally we display the isothermal lines in degree Celsius [°C]. Columns correspond to the bulk shortening state and rows to cross-section X-position.





**Figure 10.** Graph showing lateral cross-sections for two different states of bulk shortening for the geometrical evolution of model C.G-1. Cross-sections are taken in 10 km steps from X = 0 km to X = 50 km. Additionally we display the isothermal lines in degree Celsius [°C]. Columns correspond to the bulk shortening state and rows to cross-section X-position.





core of this fold was filled with half-graben sediments and progressively displaced towards the left where the basement top is horizontal (Figure 10a-e). In the region without half-graben, only a small amount of weak sediments was available to fill the fold core so that considerable amplification was inhibited (e.g. cp. Figure 10f and a), as is the case for detachment, or décollement, folding (e.g. Epard and Groshong Jr, 1993; Schmalholz et al., 2002; Butler et al., 2019). During bulk shortening, the detachment fold is displaced across the basement towards the left. The HG is also closed after 47% bulk shortening (Figure

10g-j). Similarly to the other models the weak half-graben sediments have been extruded and some of these sediments have filled the core of the detachment fold. The green marker lines in Figure 10g indicate that the fold was continuously fed by the half-graben sediments during ongoing deformation. Hence, the amplitude of the largest individual fold decreases towards the model side without a half-graben (Figure 10g,l). The strong layer initially from HB forms a thick nappe-like structure with a slightly overturned layer at the front. However, no overthrusting above the sediments from HG occured.

Compared to the reference model, model D.G-1 (Figure 11) shows only a minor basement uplift at the right boundary at 21% bulk shortening (Figure 11a-f). Instead, the bulk shortening is distributed more homogeneously throughout the basement, resulting in significant thickening also in the basement on the left model side. Because the top of the basement is weaker, the strong layers can shear-off and displace slices of the basement (e.g. Figure 11a). Further bulk shortening results in the extrusion from sediments from HG resulting in a structure resembling a fold nappe. This fold nappe is overthrust from the sediments from HB, resembling a thrust nappe (Figure 11g-l). The last deformation stage is shown for 53% bulk shortening, because

$6-7\%$ more bulk shortening is required to overthrust the sediments from HB, compared to the reference model. The strong layer from HG forming the fold nappe exhibits an overturned limb that is still in contact with the basement. Thus, this model does not form any structure similar to a thrust nappe for the sediments from HG. The passive marker symbols show gentle folding of the basement, also on the model side without half-graben (Figure 11l). In contrast, the reference model (Figure 5l)

exhibits updoming of the basement without well developed folding.

## 4.3   Nádai strain and Lode's ratio

We compute the Nádai strain $\varepsilon_s$ (Nádai and Hodge, 1963) and Lode's ratio $\nu$ (Lode, 1926) for our simulations, which are two quantities to quantify 3D finite strain. Both parameters are computed from the natural logarithm of the finite strain principal axes, $\varepsilon_1, \varepsilon_2$ and $\varepsilon_3$. The Nádai strain $\varepsilon_s$ is the octahedral shear strain and provides a non dimensional value for the strain

magnitude:

$$\varepsilon_s = \frac{1}{\sqrt{3}} \sqrt{(\varepsilon_1 - \varepsilon_2)^2 + (\varepsilon_2 - \varepsilon_3)^2 + (\varepsilon_3 - \varepsilon_1)^2} \qquad (7)$$

The Lode's ratio provides information on the strain symmetry and strain regime and can exhibit values in the interval of $[-1\ 1]$. In particular, $\nu < 0$ indicate a prolate strain ellipsoid (constrictional strain), $\nu = 0$ a prolate-oblate strain ellipsoid (plane strain) and $\nu > 0$ an oblate strain ellipsoid ellipsoid (flattening strain):


$$\nu = \frac{2\varepsilon_2 - \varepsilon_1 - \varepsilon_3}{\varepsilon_1 - \varepsilon_3} \qquad (8)$$





**Figure 11.** Graph showing lateral cross-sections for two different states of bulk shortening for the geometrical evolution of model D.G-1. Cross-sections are taken in 10 km steps from X = 0 km to X = 50 km. Additionally we display the isothermal lines in degree Celsius [°C]. Columns correspond to the bulk shortening state and rows to cross-section X-position.





We display both quantities for model A.G-1 (Figure 12) for three different cross-sections, which provides a representative image of the finite strain distribution also for the other models. The first section is taken at the maximum depth of HG (X = 0 km), the second at half of the depth (X = 25 km), and the last at the section where the half-graben disappears (X = 50 km). In general, $\varepsilon_s$ values are highest on the right side on top of the basement horst, ranging between 3.5 to 4. The extent of

this shear zone, for the same intensity, changes laterally with changing depth of HG. For example, the shear zone roots much deeper into the basin on the side without HG (cp. Figure 12a-c). Moreover, $\varepsilon_s$ outlines the fold nappe structure indicating that the fold nappe is surrounded by highly strained material. Here, maximal values of $\varepsilon_s$ increase laterally with decreasing depth of HG. Another lateral difference is the intensity of the shear zone located on the left side of the model in front of the extruded sediments. This shear zone also gains intensity in lateral direction with decreasing half-graben depth. Similarly to $\varepsilon_s$ the, Lode's

ratios $\nu$ (Figure 12d-f) pinpoint the shear zone on top of the horst and outline the fold nappe.

Overall, $\nu$ values are close to zero which indicates a plane strain deformation. The largest deviations from plane strain are located in the cross section of maximum depth of HG (Figure 12d). Furthermore, values around the fold nappe in the range of 0.5 imply a flattening regime. In contrast, the shear zone at the rear of the basement horst displays negative values that are in the order of $-0.2$ to $-0.5$, indicating a constrictional deformation.

# 5  Discussion

## 5.1  Impact of lateral geometry variations and rheological layering

Our models show the detachment of sedimentary units and their subsequent horizontal transport. Depending on the model configuration, the sedimentary units resemble fold or thrust nappes. The strain localization necessary to detach the sedimentary units and to transport them horizontally without significant internal deformation occurred without any mechanical softening

mechanism. In our models, the main cause of the strain localization is the initial geometrical configuration and the variations of mechanical strength between the model units. Strain localization and associated shear zone formation due to geometrical and strength variations has been shown with 2D numerical simulations to occur even for linear viscous material (Bauville and Schmalholz, 2017). For such strain localization no variation of effective material properties develops and it is, hence, sometimes termed kinematic strain localization, which is fundamentally different to so-called dynamic softening mechanisms in which

the effective material properties change, for example due to local heating, grain size reduction or fluid infiltration (Bauville and Schmalholz, 2017). Moreover, the formation of fold nappes by pushing ductile material against a rigid obstacle, without dynamic softening, was shown, for example, with laboratory deformation experiments (e.g. Bucher, 1956) and numerical simulations ( e.g. Peña and Catalán, 2004). Dynamic softening mechanism are most likely active in nature, but their intensity and the required deformation conditions are still contentious. Furthermore, dynamic softening mechanisms most likely intensify

the strain localization shown in our models, but our models show that, in principle, such dynamic softening mechanism are not essential to generate nappe-like structures.





**Figure 12.** Graph showing three different cross-sections of model A.G-1 displaying Nádai strain (first column) and Lode's ratio (second column). Row-wise cross-sections at X = 0, 25, 50 km, respectively.

The emplacement of the sediments from HB in the style of a thrust nappe is observed in all models with exception of model C.G.-1. The thrust nappe exhibits a laterally uniform front despite the deformation and nappe formation in HG below (Figure 4). Even in the case of model A.G-2, where the basement horst is strongly incorporated in the deformation, we do not observe large

horizontal displacement gradients in the thrust nappe. Hence, deformation of such relatively small graben systems might play only a minor role on the displacement variations along strike of fold-and-thrust belts. In our models, the uniform thrust front is the result of the initially straight boundary between basement horst and HB. Consequently, we infer that the initial large scale basin architecture plays a major role in the geometry of orogenic salients. Several studies (e.g. Thomas, 1977; Marshak et al., 1992; Boyer and Elliott, 1982) of different orogens also indicate that the sediment basin thickness is of particular importance

in the expression of salients. In this context, Macedo and Marshak (1999) investigated the effect of variable basin geometries





during bulk shortening using 3D sandbox models. Their study implies that basin-controlled salients are strongly controlled by the basin topography, that is, variations of sedimentary thickness inside the basin. There are, however, additional conditions such as the indenter shape or the direction and orientation of convergence of the colliding plates that influence the overall shape of orogens ( e.g. Laubscher, 1972; Ries, 1976; Mitra, 1997). Naturally, the investigation of such large scale boundary effects

would require a different model configuration than in our study.

Volume variations related to the lateral changes in the depth of HG are always expressed in the lateral variations of the thickness of the extruded nappe. The thinning out of the initial graben structure is reflected in a decrease of height and a decrease of length of the nappe. These lateral variations also affect the lateral amplitude variation of the major detachment fold of simulation C.G-1. The local geometry around the major detachment fold resembles a thin skinned tectonic style where most deformation is

concentrated in the cover sediment. However, much of the sediments filling the core of the detachment fold originate from HG, which is more than 10 km away from the detachment fold (Figure 11g-l). Closure of HG resulted in movement of sediments from HG into the core of the detachment fold. This result shows that detachment fold initiation and progressive evolution can be controlled by the inherited basement structure, Additionally, this finding suggest that predominately thin skinned tectonics can passively be influenced by underlying heterogeneities due to variations in the basement architecture. Nevertheless, the

expression of such detachment folds in our models also depends largely on the thickness ratio between incompetent and competent units. Our findings are, therefore, in broad agreement with previous field observations and numerical studies (e.g. Pfiffner, 1993; Jaquet et al., 2014).

## 5.2  Comparison with the Helvetic nappe system

Model A.G-1 and B.G-1 are able to reproduce several first-order structural features of the Helvetic nappe system. In addition,

we are also able to connect the two different main kinematic deformation phases of the Western Helvetics and the Eastern Helvetic Nappes. The formation of the main basal thrust originates in weak sedimentary units, comparable to the Cretaceous Palfris shales, of our Helvetic basin equivalent. Similarly to the Prabé deformation phase in the West and the contemporaneous Calanda phase in the East (e.g. Pfiffner, 2015), this basal thrust aids in the transport of a laterally uniform thrust nappe in our simulations. Given the simplifications of our model we attribute the resulting major thrust front to the formation of basal

thrust of the Wildhorn super-nappe in the West and the Glarus thrust in the East. Continuous bulk shortening leads to shear localization at the contact between the strong layer and the horst in the basin. Subsequently, the layer is detached and translated with help of the weak sediments across the basement horst and onto the half-graben, resembling a thrust sheet or nappe. We record a first noticeable deformation of the half-graben approximately 2 Ma in our simulations. This timing is in agreement with a study by Jaquet et al. (2018) which proposes a similar time interval between the onset of the Helvetic nappes and the

onset of the basal thrusting at the Morcles half-graben. Continuing, the vertical extrusion and squeezing-out of the half-graben sediments takes place simultaneous with stacking of the major thrust nappe on top at about 6 to 7 Ma. The resulting nappe stack shows major structural differences in the lateral direction of the model. This along-strike variation is comparable to the lateral structural variations in the Helvetic nappe system. At the end of our simulations we record approximately 10 Ma for the





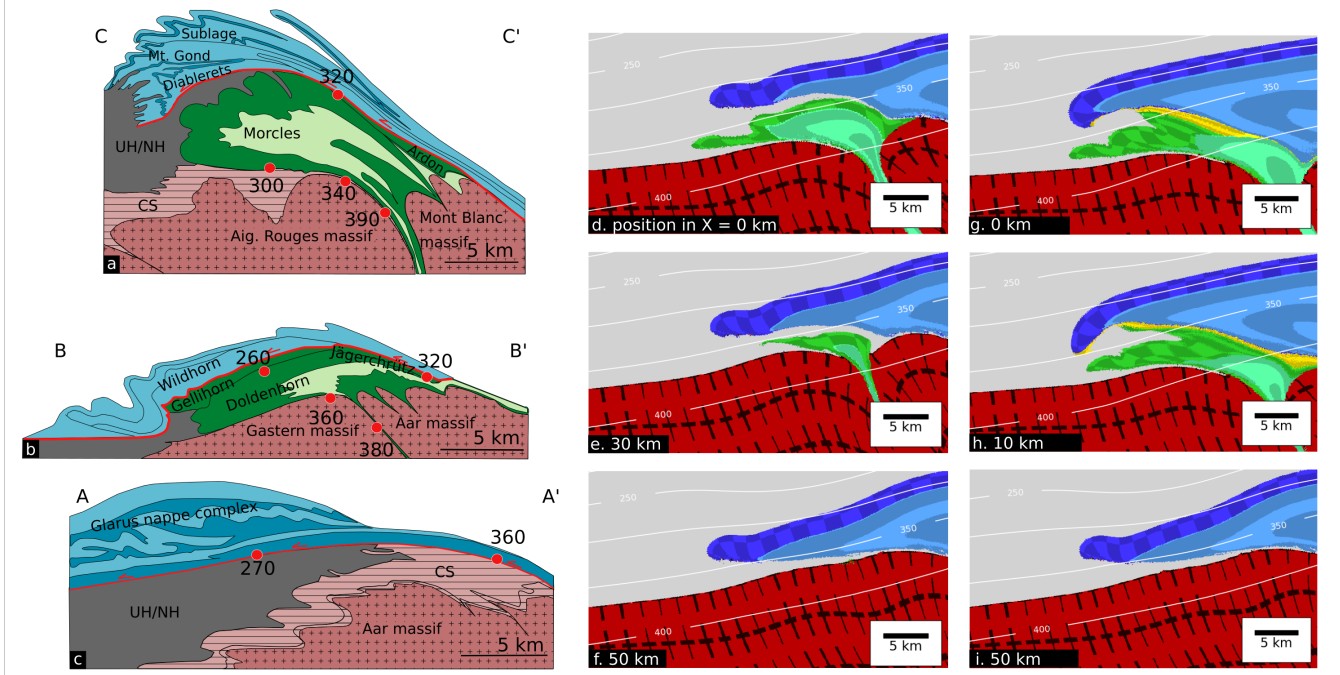

**Figure 13.** Comparison between the geological cross-sections of the Morcles nappe, Doldenhorn nappe and Glarus nappe complex with sections taken from our numerical simulations. Columnwise(left to right) 1.column: geological-cross sections, 2.column: sections from model A.G-1, and 3.column: sections from model D.G-1. Temperatures taken from Ebert et al. (2008). UH/NH = Ultrahelvetics/North Helvetics. CS = Cover sediments.

complete process of nappe stack formation. This timing is in broad agreement with studies of the Morcles nappe complex (e.g.

Kirschner et al., 1996; Boutoux et al., 2016) that suggest an emplacement duration of 10 to 15 Ma.

Figure 13 shows a comparison between geological cross-sections taken along-strike of the Helvetic nappe systems with selected zoomed in numerical cross-sections in lateral direction of our model. The first cross-section a) of model A.G-1 (Figure 13d,e,f) shows our equivalent of the Morcles nappe at the maximum depth of the half-graben. The result here is similar to 2D numerical studies by Bauville and Schmalholz (2015) who investigated fold nappe formation and nappe stack formation also in application

to the Helvetic nappe system, respectively. To first order we reproduce an extruded fold nappe with a shape comparable to the Morlces nappe, which means that its the length only slightly exceeds its height. The strong layer is still connected to the adjacent basement horst and the deformed internal weak units, highlighted by the green passive marker lines, resembles a recumbent fold. The inner part of the fold nappe roots into a steep cusp, analogues to the Chamonix zone between the Aiguilles-Rouges and Mont Blanc massif (Figure 13a). Moreover, we find significant amount of material from the overburden between the thrust

nappe and the fold nappe, but also below the fold nappe. The material between the two nappes resembles the Ultrahelvetics in the geological cross-sections, whereas the material below the fold nappe corresponds to undeformed cover-sequences of the basement. Turning now to the Morcles equivalent of model B.G-1 (Figure 13g) we find a significant different structure of




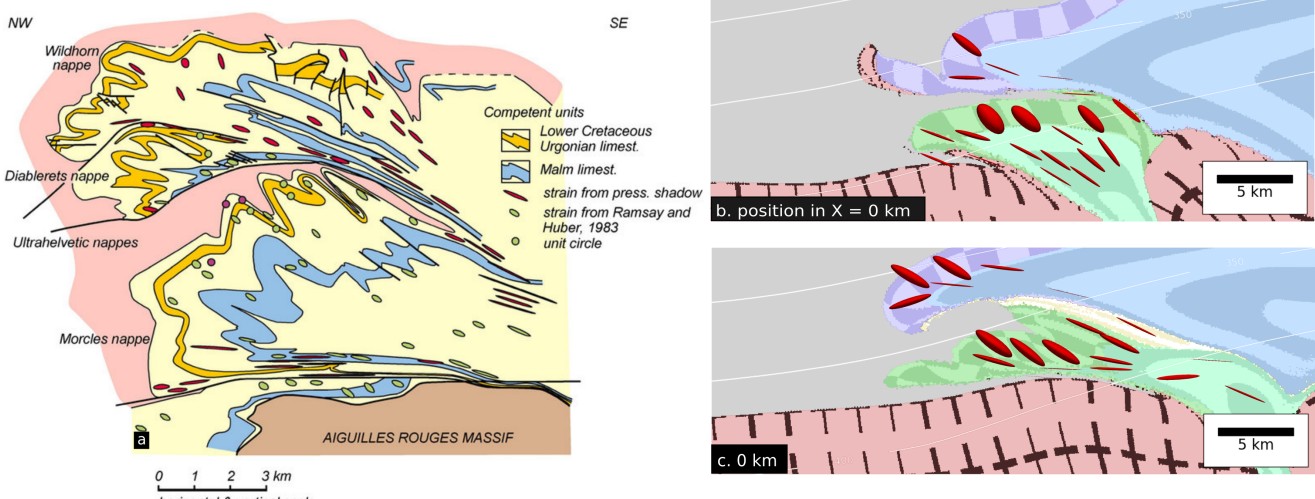

**Figure 14.** Comparison between finite strain from the Morcles nappe (taken fromBastida et al. (2014); after Casey and Dietrich (1997) modified from Dietrich and Casey (1989), green strain ellipses from Ramsay and Huber (1987)) and selected finite strain ellipses from model A.G-1 and model D.G-1.

the extruded fold nappe. There are several major differences that are connected to the initial mechanical layering inside the half-graben. First, the fold nappe shows a far greater length to height aspect ratio than the natural analogue. Secondly, the
strong normal limb is not in contact with the basement horst. Due to the continuous strong layer the structure shows greater shearing at the contact between the fold and thrust nappe.

Proceeding with cross-section e) of our numerical model A.G-1 (Figure 13e), we find similarities with the geological cross-section of the Doldenhorn nappe (Figure 13b). The extruded nappe shows similar aspect ratio and shape like the Doldenhorn nappe with the length significantly exceeding the thickness. However, cross-section h) of model B.G-1 (Figure 13h) provides
an even better match with the structure of the Doldenhorn nappe. The resulting fold nappe shows a comparable shape and size. Due to the initial rheological layering this model forms a thin thrust sheet on top of the fold nappe. We interpret this thrust sheet as an analogue to the Gellihorn and Jägerchrütz nappes, which are essentially thin thrust sheets on top of the Doldenhorn nappe (Pfiffner et al., 2011).

Finally the numerical cross-sections without half-graben (Figure 13f,i) exhibit a similar displacement and shape for both of our
models. The finding suggests that underlying nappe formation in HG and minor variations in the vertical rheological layering has a negligible effect on the horizontal displacement parallel to the shortening direction along strike of the major thrust sheet. To first order, this structure is comparable to the Glarus nappe complex displayed in Figure 13c.

Additionally, we also find a first order agreement when comparing the finite strain pattern between field measurements from the Morcles nappe and our numerically calculated finite strain (Figure 14). Generally, the aspect ratio of the strain ellipses increases
towards the bottom and root zone of the nappe, indicating significant shearing. Furthermore, we observe less deformed ellipses



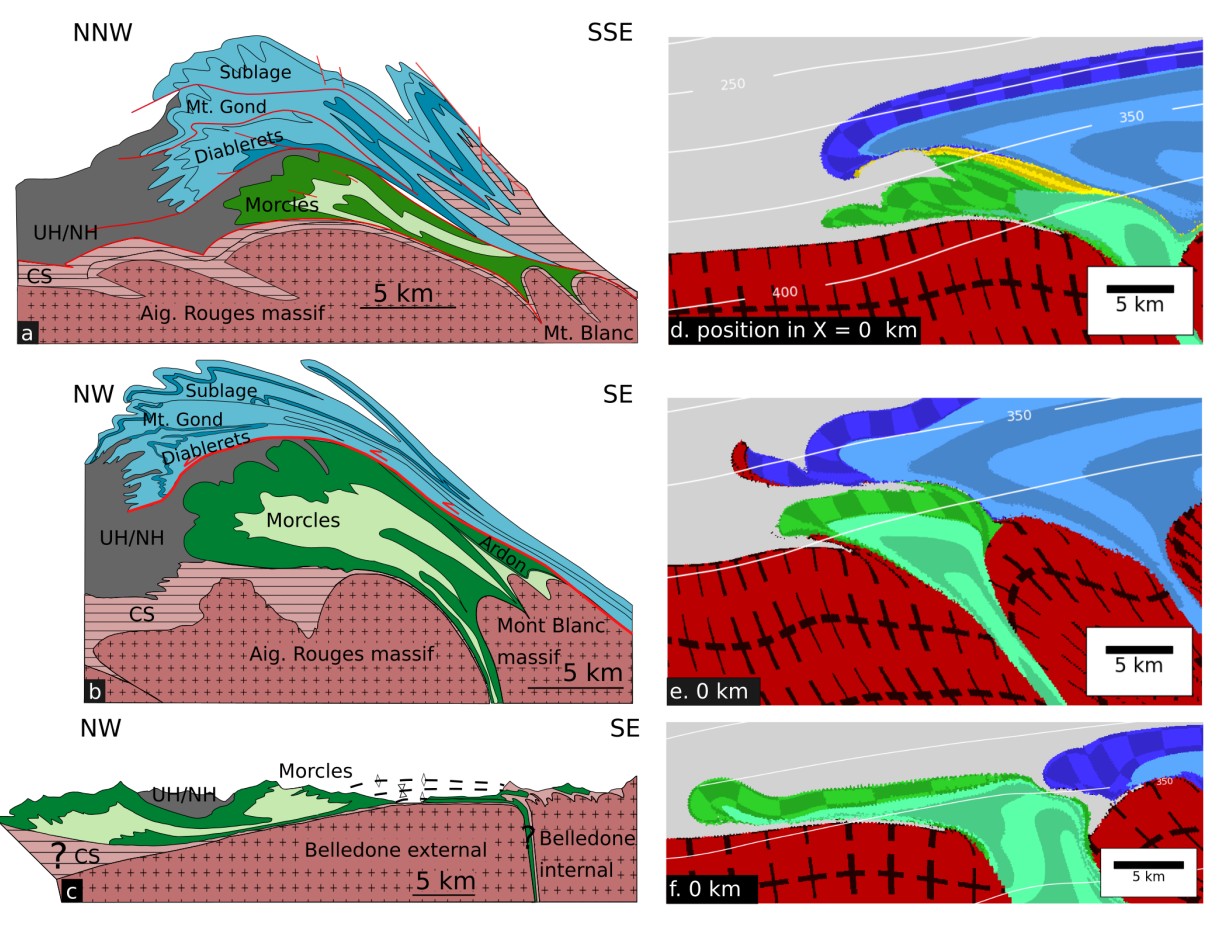

**Figure 15.** Comparison between selected numerical cross-sections and simplified geological cross-sections. Row-wise(top to bottom) a) Simplified Morcles section modified after Pfiffner (2015) (indicated in the map of Figure 1 with the dashed profile line) and d) section from model D.G-1. b) Simplified Morcles modified section after Escher et al. (1993) and e) cross-section model F.G-1. c) Simplified section modified after Epard (1990) and f) section from model A.G-2. UH/NH = Ultrahelvetics/North Helvetics. CS = Cover sediments.

in the top, normal limb of our Morcles equivalent with ellipses of higher aspect ratios (Figure 14b, d) near the contact between the fold nappe and the thrust nappe. Figure 14b shows a better match for the finite strain pattern in the upper limb, whereas Figure 14c displays a better fit for the contact zone. We also observe a subhorizontal orientation of the major finite strain axis towards the root zone of the nappe (Figure 14c). Numerical finite strain computation is a useful tool to compare numerical
models with natural observations. The comparison can efficiently be used to recognize where the numerical model requires adaptation to the real data.

However, certain first order features of the Helvetic nappe system we could not reproduce with our models. The first shortcoming is the protrusion and shearing out of the strong layer of the half-graben. In 3D view this process is expressed as a thin carpet





in front of the fold nappe (e.g. Figure 4c,f), resembling a nose-like feature in the 2D cross-sections. Shearing out of the layer
inhibits the formation of an outer recumbent limb as observed in the Morcles nappe (Ramsay et al., 1983). Here, our results are
in contrast with previous 2D thermo-mechanical numerical modelling results by Bauville and Schmalholz (2015), who success-
fully simulated the formation of a recumbent fold limb during half-graben inversion. One of the main reasons for this deviation
lies in the viscosity ratio ($\eta_R$) between overlying strong layer ($\eta_L$) and the half-graben infill ($\eta_I$). In the simulations of Bauville
and Schmalholz (2015), both the layer and the infill have the same viscosity ($\eta_R = 1$), whereas our simulations, using different
creep flow laws for each units, show that the infill viscosity $\eta_I$ can be up to three orders of magnitude smaller than the strong
layer viscosity $\eta_L$ ($\eta_R = 1000$). Secondly, temperature estimates from several authors (e.g. Kirschner et al., 1999; Herwegh
and Pfiffner, 2005; Ebert et al., 2007b) indicate lower peak temperature conditions along the different basal nappe contacts in
comparison with our simulations. For example, Ebert et al. (2008) reports increasing temperatures of $270°C$ to $390°C$ from
the front of the Morcles nappe to its root. Here we find a close match with root zone temperatures, whereas our front part
of the fold nappe exhibits temperatures in the range of $370°C$ (e.g. Figure 13d). Similar temperature comparisons between
our Doldenhorn and Glarus thrust complex equivalents show the same trend (see Figure 13). Moreover, the cross-sections of
model B.G-1 exhibit slightly higher temperatures inside the core of the fold nappe, in consequence of the significantly thicker
thrust nappe above. In summary our models show a good match for the root zone temperatures, but higher temperatures at the
front of the nappe complexes compared to natural observations. Ebert et al., 2008 suggests a horizontal temperature gradient of
approximately $6°C/km$ along the nappe interface for all three cross-sections. We only register a horizontal gradient of $2°C/km$
along the basal thrust surfaces. Additional constraints are given by Leloup et al. (2005) and Boutoux et al. (2016) who indicate
peak metamorphic temperatures of $320°C$ for the Aiguilles-Rouges massif and $400°C$ for the Mont Blanc massif. Despite these
discrepancies our model conforms to the temperature trends of natural observations, showing increasing temperatures from
the front to the root zones of the nappes. Further improvements could be made by adjusting the initial geothermal gradient
or by modifying the initial geometrical configuration. For the latter case, primarily, Nibourel et al. (2018) demonstrate that
the Aar massif experienced a $10 - 15°$ southwards dip in relation to the isotherms. Essentially such a configuration would,
for example, place the half-graben system further up in the isotherms. This adjustment would lead to cooler temperatures in
the front of the nappe and greater horizontal temperature gradients during and after nappe emplacement. In terms of large
scale structural components our simulations lack the formation of parasitic folds, smaller imbricate thrusts or the detachment
between different levels inside the major thrust complex. Modelling of such smaller-scale features would require (i) a higher
numerical resolution in combination with (ii) a drastically more complex rheologically layering. In addition, field observations
indicate the existence of several large scale shear zones in the Mont Blanc and Aar massif. Our models do not reproduce such
features, mainly because of our mechanically homogeneous basement unit which represents a large simplification compared
to the natural complexities. There are, for example, several studies regarding the Aar massif that correlate the formation of
ductile and brittle shear zones to inherited pre-alpine heterogeneities ( e.g. Berger et al., 2017; Mair et al., 2018). In particular,
structures such as foliations, mafic dykes or folds present mechanical anisotropies that can culminate in localized strain and
shear zones (e.g. Bell, 1978; Carreras et al., 2010; Herwegh et al., 2017; Wehrens et al., 2017).





Finally, our model does not fully resolve the large scale kinematics and intricate deformational history of the Helvetic nappe system. For example, reconstructions between the SW and the NE of the Helvetics show a differential horizontal displacement

of the thrust nappes of up to 50 km ( e.g. Pfiffner, 2015). Here, variations are likely due to the initial basement geometry and convergence of subducting plate e.g. obliquity of the plate which have not been taken into account in our model.

## 5.3 Comparing geological with modelled cross sections: The Morcles nappe

The method of cross-sectional balancing and reconstruction is one of the major tools to understand the evolution of fold-and-thrust belts on different structural scales (e.g. Dahlstrom, 1969; Price, 1981; Baby et al., 1992; Massoli et al., 2006; Alavi,

2007). Even though some authors suggest an underlying bias of such techniques due to different simplifications (Butler et al., 2019), geological cross-sections themselves are indispensable. In addition, they are useful for comparisons with numerical models that aim to decipher the mechanical processes and material properties that lead to the formation of geological structures. However, caution is advised when comparing sections of numerical models with geological cross section. The large three-dimensional variability in geological structures and structures generated in numerical models such as in this study should

be a warning sign when trying to fit one observation with one model. Hence, in this section we show and compare three different geological cross sections of the Morcles nappe with our numerical results (Figure 15). The cross sections are from different locations along strike of the Morcles nappe and highlight, together with our numerical cross section, the complexity of geological reconstruction. We start with a comparison of a geological cross section of the Morcles nappe near the Sanetschpass and a section of model B.G-1 (Figure 15a and d). Here, the geological cross section shows a relatively thin Morcles nappe

with a sheared lower and upper limb. This observation is similar to our cross section which exhibits an elongated sheared fold nappe with sheared lower and upper limbs. Moreover, in this model we observe a significant deformation of the basement at the contact of the cover sediments which is in agreement with microstructural observations that suggest a brittle-ductile emplacement of the Morcles nappe (Ebert et al., 2007a; Austin et al., 2008). Figure 15b shows a Morcles cross section of Escher et al. (1993) from further Southwest in the nappe system. In this case, model D.G-1 (Figure 15e) provides a better first

order fit than B.G-1. Our fold nappe exhibits a strongly sheared recumbent limb that still reaches into the root zone of the nappe. Furthermore, the strong layer is still connected to the basement horst and is in contact with the thrust nappe where it is sheared, displaying a similar saw-tooth shape as observed in the geological cross section. Also, the overall aspect ratio is closer to the geological cross-section than in model B.G-1. In contrast, both of the previous numerical sections do not match the observations for the Morcles nappe even further Southwest in France near Megève (Figure 15c). Here, the sediments are

rather squeezed out of the half-graben due to a greater thickness of shales instead of forming a recumbent limb (Epard, 1990). Therefore, we suggest that a cross-section of model A.G-2 Figure (14f) shows a closer match to the geological structure. In summary, different cross sections of the Morcles nappe can be compared to different numerical models with different initial conditions regarding the geometry or rheology. Hence, it is a challenge to capture the evolution of 3D geological structures with a single 3D model, because of the geometrical and rheological uncertainties. Further, this also implies that one has to





be even more cautious in the application of a single 2D model to the formation of 3D fold-and-thrust belts and geological structures in general.

# 6 Conclusions

The presented 3D numerical simulations show the formation of a fold-and-thrust belt resulting from the wedge-type deformation of the upper crustal region of a passive margin. The deformed crustal region is characterized by a half-graben with
laterally varying thickness and a horst separating the half-graben from a laterally homogeneous basin. The numerical simulations show the formation of tectonic nappes with horizontal displacement of several tens of kilometers and with geometries ranging from fold nappe to thrust nappe. The formation of the sedimentary nappes results from the shearing-off and detachment of sedimentary units form the half-graben, horst and basin. Nappe detachment, transport and stacking occur for standard viscoplastic rheological models without any applied rheological, or dynamic, softening mechanisms. Nappe formation and their
geometry are controlled by the initial basement geometry and the strength contrast between basement and cover sediments and the strength contrast within the sediments. The results indicate the fundamental importance of tectonically inherited structures on the evolution of fold-and-thrust belts. Consequently, the results emphasize the importance of geological field work and reconstructions of the initial geological situation before fold-and-thrust belt formation.

Modelled nappe-like structures generated from sediments from the half-graben with laterally varying depth show that the nappe
geometry strongly depends on the amount of sediments available for nappe formation. The calculation of 3D finite strain shows that the deformation during the formation of nappes originating from the half-graben largely deviates from a 2D plane strain deformation. Nappes originating from the laterally homogeneous basin show a more or less laterally straight displacement front indicating that the laterally heterogeneous deformation during nappe formation in the half-graben region does not considerably affect the overthrusting nappe.

The applied strength of the basement has a strong control on the resulting nappe geometry. A relatively stronger basement favors the formation of nappes resembling a thrust nappe while reduced basement strength (due to a stress cut-off at 40 MPa) favors the formation of fold nappes.

The initial model configuration was based on geological reconstructions of the Helvetic nappe system. The modelled 3D structures show several first-order similarities with this nappe system: (1) Formation of a nappe resulting from the closure
of a half-graben. Depending on the model configuration, this nappe can be more similar to a fold or a thrust nappe. The modelled nappes are applicable to the Morcles and Doldenhorn nappes of the Helvetic nappe system. (2) Formation of a laterally homogeneous thrust nappe which overthrusts and is stacked above the underlying nappe resulting from half-graben closure. This nappe is applicable to the Wildhorn and Glarus nappe. (3) The detachment of minor sediment units originally located on the horst and their emplacement between the upper thrust nappe and the underlying nappe from the half-graben.
These minor nappes are applicable to the Ardon, Jägerchrütz and Gellihorn nappes. (4) The entrapment of weak sediments, which were originally situated structurally above the sediments eventually forming nappes, between the two major nappes.





These entrapped sediments are applicable to the Ultrahelvetic units. (5) The modelled temperatures, temperature gradients and finite strain gradients are in overall agreement with data from the Helvetic nappe system.

# Acknowledgements

This work was supported by SNF grant No. 200020-149380 and the University of Lausanne. Moreover, this work was supported by a grant from the Swiss National Supercomputing Centre (CSCS) under project ID s785. In addition we thank Ludovic Räss and Philippe Logean for supporting us in the utilization of the Octopus Super Computing Cluster at the University of Lausanne.





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
