# Peer review of "Control of 3D tectonic inheritance on fold-and-thrust belts: insights from 3D numerical models and application to the Helvetic nappe system"

_Solid Earth, 2019_

## Referee Comment (RC1) · O. Adrian Pfiffner (Referee) · 19 Dec 2019

Review of Spitz et al. se-2019-173

General comment

All in all this manuscript addresses an interesting topic in a clear way. My comments in this review are meant to improve the manuscript.

Abstract is too long. The first three sentences is not what you found. They should be in the aims part of the introduction. The important statements are found from line 10 to 24.

1 Introduction is OK

2 Geological overview is OK, but a few corrections needed (see specific comments).

3.1 Numerical method OK, but the mathematical part is not my field of experetise. Fig. 3 is hard to read (dark red color hinders 3D visualization, model types are too small to read) 3.2 Model configuration: description is OK. But I think that the simple geometry of a graben getting narrower linearly is not the best solution. It might well be that the passive margin is fragmented as shown by Trümpy for the Early Jurassic in eastern Switzerland with NS-striking synsedimentary faults (transformlike). Considering the en echelon pattern of the hinge lines of the Morcles and Doldenhorn nappes it seems more logical to use a fragmented basin (an offset placing the northern graben bounding fault farther South for the Doldenhorn area). Fragmented passive margins are more the rule than the exception (see papers by the Manatschal group in the Austroalpine units, the Pyrenees and the Atlantic off Portugal).

4. Results 4.1 Section on 3D Model evolution is OK, Fig. 4 is a nice summary graph, which would fit into section 5.1 4.2 Section on 2D numerical cross-sections. Cross-sections are clearly presented, but the wealth of data seems somehow "too much". The figures could be condensed by showing two cross-sections, at x=0 and x=40 km. The intermediate states show basically the same features and do not exhibit significant changes. The text would of course need to be modified (shortened) accordingly. The full detail could be provided as supplementary material. 4.3 Nadai strain and lode's ratio is OK, but I asked myself what you wanted to extract from this information. The deviation from plane strain has been a problem in structural geology that never has been satisfactorily been resolved. Your 3D modeling could give us some clues. But even in the discussion and conclusion you do not take advantage of the data the model provides.

5 Discussion 5.1 Impact of lateral geometry variation is OK. Fig. 4 would be well (better) placed in this section! 5.2 Comparison with the Helvetic nappes is OK, with

a few corrections that need to be included (see specific comments). 5.3 Comparing geological with modeled cross-sections, Morcles nappe. This section has important flaws (see specific comments) such that I tend to suggest deleting it.

6 Conclusions are a bit lengthy. They contain statements that belong to the abstract. True conclusions (what was learned from the research) formulated to the point. Some of the language is a bit cumbersome (see specific comments)

Specific comments

55 You mention analogue models but do not discuss them at all later. If you wish to make reference to analogue models you need to add a few explanations with references.

123 It is important to note here that it is the Early Jurassic basin that plays the major role in the development of the internal structure of the nappe. This basin is restricted to the area west of the Aar massif. In eastern Switzerland the Early Jurassic basin is restricted to the area south of the Aar massif. "North Helvetic basin" is misleading as term (it is also used in conjunction with flysch basins).

134 see 123

136 which carbonates are your referring to by saying "in between"? The carbonates are of Late Jurassic and Cretaceous age and rest on the marly-shaly Early and Middle Jurassic sediments.

137 I disagree with these differences: for Morcles shearing at its base is really prominent as well and the internal folds of the Doldenhorn are isoclinal in part, and the length of the overturned limbs are comparable.

150 The Rawil depression is not a "topographic" feature (Wildhorn and Wildstrubel are among the higher peaks). It is a structural depression.

152 The Early Jurassic basin is not proven to be continuous. As a matter of fact the

hinge lines of Doldenhorn and Morcles are clearly not lined up and can therefore not be correlated. These hinge lines are most likely controlled by the basin architecture, which is a primary target for this study. By saying that the basin is continuous your are making an assumption that you want to investigate by your study. If the hinge lines of Morcles and Doldenhorn reflect the orientation of the northern basin border then this border must have a jog. Some people (e.g. Burkhard) explained this jog as a NS-striking strike-slip fault. But in reality we do not have any data on this. The seismic data of NRP20 are inconclusive on this.

154 The statement "absence of significant nappes in the Infrahelvetic complex" does not correspond to reality: there are three major nappes (Kaminspitz, Calanda and Tschep), all of which have significant displacements. But what they lack are recumbent folds, a fact that reflects the absence of an Early Jurassic basin and the Middle Jurassic sediments being very thin in comparision to the Late Jurassic and Cretaceous carbonate sequences.

155 The Doldenhorn nappe is much much closer in style to the Morcles nappe; it displayes long inverted limbs which are absent in the Glarus nappe complex in eastern Switzerland.

160-162 This interpretation is contested sharply by Pfiffner et al. (2011).

509 The Chamonix-zone does not show a synclinal structure in nature. The Mesozoic sediments show a consistent younging from the Mont Blanc massif to the NW. The youngest sediments then but against the Aiguilles Rouges massif's basement. This is clearly visible form the structural maps 1:100'000 that you cite earlier on (Pfiffner et al. 2010).

498 This statement about the Helvetic nappes needs a reference.

512/13 Fig 14 The intention of this figure is much appreciated. I have some worries though if strain ellipses determined from pressure shadows are directly compared to

strain determined from deformed oolites. And what are the contributions to the figure by Bastida, Dietrich & Casey, Casey & Dietrich? Strains or cross-sectional geometry? Or are all the strain data from Ramsay & Huber? And I miss the effect of the Permo-carboniferous graben in the Aiguilles Rouges massif (it is partially inverted and folds the Morcles thrust above.

578-605 It is no surprise that the three cross-sections chosen from along strike give different results. Cross-section shown in Fig. 15a is from the Rawil depression where the Morcles-nappe is deeply buried in the subsurface and thus drawn by projection only. The top basement beneath is constrained by seismic data of NRP20 and thus explains somewhat the reduced thickness of the nappe. However, I never put a name to the basement uplifts because of the uncertainty involved and urge the authors to delete them. One could equally well put the names of the Gastern and Aar massif in their place. The cross-section is more reliable for the Wildhorn nappe since this nappe outcrops along the trace of the cross section. The cross section shown in Fig. 15c shows a completely different nappe – and I doubt very much that it should be called Morcles nappe. In fact the Morcles nappe in the type locality "Dent de Morcles" displays hinge lines of internal folds that climb westwards over the Aiguilles Rouges massif, crossing its crest line and then plunge towards the SW beneath the Chablais and the Chaînes Subalpines thrust sheet. The structures shown in Fig. 15c are merely in the same structural position relative to the Chaînes Subalpines thrust sheet. The uncertainty emanating from the construction of (balanced) cross-sections could be extracted from the numerous cross-sections drawn along the trace of the cross-section shown in Fig. 15b. One of the main reason for divergent solutions is the observation that the lower limb is more horizontal whereas the upper limb plunges with 30° to the NE (see discussion in Pfiffner 1993, a reference referred to in the manuscript). There aren't many cross-sections constructed along curved hinge lines as is necessary in this situation. The one shown in Pfiffner (1993) is based on the construction of Langenberg et al. (1987) who did use curved hinge lines. The major effect of the differing plunges is the thickness of the Morcles nappe. Curved hinge lines yield a thickness of ca. 5 km, the

cross-section by Escher et al. (1993) used in Fig. 15B suggests 7 km.

609-613 Does not present a conclusion. For me one important conclusion is the next following sentence (Nappe detachment, transport . . ..)

618-619 The importance of fieldwork in such a scenario is common sense.

623 This is the place that screams for a statement on the nature of the strain in and out of the cross-sectional planes.

628-638 These are findings that should go into the abstract.

Technical corrections

18 French-Swiss Alps is not a common denomination. I suppose you wish to include the Haute Savoie part of France. I suggest "Central Alps of France and Switzerland"

74, 108 see 18

76-77 "laterally" instead of "along the lateral direction"?

111 I suggest "Glarus nappe complex of eastern Switzerland"

148 Diablerets

326 2D numerical cross sections: why specify "numerical" here? All is numerical. And wouldn't "thrust sheet" be a better term than "thrust nappe", particularly as it is opposed to "fold nappe"?

496-497 suggestion: The resulting model nappe stack shows laterally major structural differences.

515 It would be better to formulate what is observed, and not what is not observed (contact to basement)

548 report (not reports)

549 frontal part

551 Doldenhorn and Glarus nappes (or Doldenhorn nappe and Glarus nappe complex)

554 suggest

563 start a new paragraph with "In terms of…..

624 modeled by a stress cut-off at 40 MPa (instead of "due to")

Adrian Pfiffner

---

## Referee Comment (RC2) · Frédéric Mouthereau (Referee) · 21 Jan 2020

Control of 3D tectonic inheritance on fold-and-thrust belts: insights from 3D numerical models and application to the Helvetic nappe system by Spitz et al. Submitted for publication to Solid Earth

General comments The paper uses 3D numerical thermo-mechanical modelling of a heterogenous rifted margin with variable basement/sediments and structures assuming viscous-plastic laws. Results are then compared to deformation observations along 3 sections of the external Swiss-French Alps that have reached temperature conditions of 250-380°C. The model setup is intended to reproduce the initial architecture of grabens

of the European margin (proximal part) with different sediment thickness and geometry of basins varying along-strike. A V-shaped North-Helvetic basin is assumed. One main implications of such modelling approach is that for sedimentary units to be detached above the basement and form thrust nappes (little internal deformation) no mechanical softening is required. Strain localization results from the geometry and strength variations, which conditions are likely met in many mountain belts. The overall modelling presentation and results are well written and quite easy to follow. I am only concern about erosion that is not modelled (see below). Results are sounding and the applicaiton to the Alps relevant, but please consider say some words about the choice of having discarded the role of erosion. I only suggest to consider reorganising/rewriting the Introduction and the Discussion (Section 5.3).

Specific comments Introduction Lines#44-50: This paragraph is very specific (i.e. only viscous mechanism are addressed) relative to the rest of the introduction. I suggest to move them after lines#55-61 where the authors present older studies with more general mechanical behavior. In addition, in classical model of FTBs (like later in the intro) a major decoupling level (low friction or linearly viscous like salt) if often assumed. The high contrast between basal and internal strength allows thrust nappe formation. This best applies to external zones when pressure and temperature (well below 300°C) does not modify the original rheological layering. In the case of shear zones the authors study here (intermediate domains of tectonic burial) this is very different because such weak layers where deformation localizes is not prescribed so a "self-localization" process is required. The introduction should better emphasize the differences between the end-member approaches. One "cold" frictional classical approach of Coulomb-type thrust wedges vs one "hot" viscous-plastic approach of ductile nappe stacking. They are both valid and should co-exist, depending which part of the orogen you are dealing with. Here we are clearly more interested in ductile-type fold-thrust belts.

Methods. The authors should indicate why erosion is not relevant or not taken into account in their modelling approach. Because erosion is not considered here rocks do not

cool during deformation, thrusting/folding and the crust is thickening. Therefore they are always in the ductile field. This may be valid but should be clearly presented especially the authors are dealing with the most recent Miocene sequence of shortening in Alps and erosion is a major factor in orogens during these late stages.

Discussion The authors are able to reproduce thrust nappes (little internal deformation) of sedimentary units detached above the basement without mechanical softening. This is important. "Kinematic" strain localization is the result of the geometry and internal strength variations.

Lines#453-456: Section 5.1. This part raises a very interesting point. The impact of this result is that the initial rheological layering and configuration of these layers suffice to produce thrust nappes. I am wondering how this could extend to the application of "static" models of brittle thrust wedges (e.g. Dahlen et al., 1990) with no mechanical softening or more dynamic ones including friction (Ruh et al., 2012). These models indeed assume that mechanical properties do not change with time and incremental displacement. Lines#463-470: Thickness does matter to explain salients and recesses in FTBs but these peculiar structure may also reflect the occurrence of laterally discontinuous decollement levels (changes in lithology and thickness - for viscous décollement - also play an important role); in other words not only changes in the overburden thickness are relevant.

The Section 5.3 presents a discussion on our attempts to compare geological sections with numerical models. I had hard time to follow the reasoning here. First the authors introduce the balanced cross-section approach which can only scarcely be applied here because of the dominant ductile behavior of the HN; thickening/thinning of layers (and lengths variations) during deformation and out-of-plane deformation preclude the use of 2D balancing techniques. 3D balancing techniques with volume balancing could do it. I think the value of this section is not the comparison with balanced cross-sections. I would suggest to focus on the 3D aspects of their numerical approach and explain how the structural complexities seen in 3D modelling could be simply explained

by lateral variations in the original structure rather than by mechanical complexities mechanics. This is important when searching for mechanical interpretation of lateral changes in cross-sections (balanced or not !). Maybe is what the authors intended to do here but could not clearly see it from my reading.

Other technical corrections are in the pdf attached.

Please also note the supplement to this comment:
https://www.solid-earth-discuss.net/se-2019-173/se-2019-173-RC2-supplement.pdf

[Figure]

**Supplement:**

[revised manuscript text omitted]

---

## Author Comment (AC1) · 4 Mar 2020

Author's response to
All in all this manuscript addresses an interesting topic in a clear way. My comments in this review are meant to improve the manuscript.

Abstract is too long. The first three sentences is not what you found. They should be in the aims part of the introduction. The important statements are found from line 10 to 24.

We have shortened the abstract.

1 Introduction is OK

2 Geological overview is OK, but a few corrections needed (see specific comments).

3.1 Numerical method OK, but the mathematical part is not my field of experetise. Fig. 3 is hard to read (dark red color hinders 3D visualization, model types are too small to read) 3.2 Model configuration: description is OK. But I think that the simple geometry of a graben getting narrower linearly is not the best solution. It might well be that the passive margin is fragmented as shown by Trümpy for the Early Jurassic in eastern Switzerland with NS-striking synsedimentary faults (transformlike). Considering the en echelon pattern of the hinge lines of the Morcles and Doldenhorn nappes it seems more logical to use a fragmented basin (an offset placing the northern graben bounding fault farther South for the Doldenhorn area). Fragmented passive margins are more the rule than the exception (see papers by the Manatschal group in the Austroalpine units, the Pyrenees and the Atlantic off Portugal).

We modified figure 3 a bit and increased, amongst others, the graphics in panel c). However, we kept the original colors. We kept the linearly narrowing graben for simplicity, but we discuss that the natural situation might have been more complex. The aim of our modelling is to understand and quantify the first-order effects of laterally varying graben structures, which is the reason why we tried to keep the initial geometry as simple as possible.

4 Results 4.1 Section on 3D Model evolution is OK, Fig. 4 is a nice summary graph, which would fit into section 5.1 4.2 Section on 2D numerical cross-sections. Cross-sections are clearly presented, but the wealth of data seems somehow "too much". The figures could be condensed by showing two cross-sections, at x=0 and x=40 km. The intermediate states show basically the same features and do not exhibit significant changes. The text would of course need to be modified (shortened) accordingly. The full detail could be provided as supplementary material.

We kept all the original panels, because we want to show in detail the lateral variations in the model. The fact that these lateral variations are sometimes not significant is in our opinion also an important result. Since we have the possibility to show these results, we prefer to present our results in the figures, rather than describing the results by words.

4.3 Nadai strain and lode's ratio is OK, but I asked myself what you wanted to extract from this information. The deviation from plane strain has been a problem in structural geology that never has been satisfactorily been resolved. Your 3D modeling could give us some clues. But even in the discussion and conclusion you do not take advantage of the data the model provides.

We agree that a discussion and analysis of the deviation from plane strain is not done in detail in our study. We focus here on the major, first-order results, which show that significant deviation from plane strain only occurs in narrow zones around the nappe boundaries.

5 Discussion 5.1 Impact of lateral geometry variation is OK. Fig. 4 would be well (better) placed in this section! 5.2 Comparison with the Helvetic nappes is OK, witha few corrections that need to be included (see specific comments). 5.3 Comparing geological with modeled cross-sections, Morcles nappe. This section has important flaws (see specific comments) such that I tend to suggest deleting it.

A main aim of our modelling study is the application to natural fold-and-thrust belts, in particular to the Helvetic nappe system, so that we keep this section. We are aware that there are different, and partly opposing, geological interpretations, in particular concerning the tectonic relationship between the Morcles nappe and the Chaînes sub-alpines in the considered region. We extended our discussion and tried to clarify our interpretation. We also extended the discussion concerning the different cross sections. Geological cross sections can be considered as geometrical models, which depend on several assumptions such as choosing an appropriate projection method. However, such sections represent the main, summarized information from geological field work and represent one of the few "data sets" which can be used for the comparison with numerical models. Therefore, we keep our discussion on the different sections, but discuss their uncertainties and alternative interpretations.

6 Conclusions are a bit lengthy. They contain statements that belong to the abstract. True conclusions (what was learned from the research) formulated to the point. Some of the language is a bit cumbersome (see specific comments)

We modified the Conclusions and tried to make them more concise.

Specific comments

55 You mention analogue models but do not discuss them at all later. If you wish to make reference to analogue models you need to add a few explanations with references.

Since our manuscript is already quite long and entirely focuses on numerical modelling, we do not want to discuss and explain the analogue results. We, therefore, deleted the references to the analogue models.

123 It is important to note here that it is the Early Jurassic basin that plays the major role in the development of the internal structure of the nappe. This basin is restricted to the area west of the Aar massif. In eastern Switzerland the Early Jurassic basin is

restricted to the area south of the Aar massif. "North Helvetic basin" is misleading as term (it is also used in conjunction with flysch basins).

We mention now explicitly that the term North Helvetic basin refers in our study to the Jurassic rift basin, and not to the much younger North Helvetic flysch basin that is part of the North Alpine foreland basin.

134 see 123

See our reply to 123.

136 which carbonates are your referring to by saying "in between"? The carbonates are of Late Jurassic and Cretaceous age and rest on the marly-shaly Early and Middle Jurassic sediments.

We removed the "in between" confusion. We omit a detailed description of the lithologies here, as it is not essential for our study.

137 I disagree with these differences: for Morcles shearing at its base is really prominent as well and the internal folds of the Doldenhorn are isoclinal in part, and the length of the overturned limbs are comparable.

We agree that the current sentence is not clear enough. We removed this sentence, since a clarification of this statement would need a significantly longer and detailed explanation, which is out of the scope of the study. The statement is also not important for our study.

150 The Rawil depression is not a "topographic" feature (Wildhorn and Wildstrubel are among the higher peaks). It is a structural depression.

We agree and replaced 'Due to the topographic Rawil depression' with 'Due to the structural Rawil depression'.

152 The Early Jurassic basin is not proven to be continuous. As a matter of fact the hinge lines of Doldenhorn and Morcles are clearly not lined up and can therefore not

be correlated. These hinge lines are most likely controlled by the basin architecture, which is a primary target for this study. By saying that the basin is continuous your are making an assumption that you want to investigate by your study. If the hinge lines of Morcles and Doldenhorn reflect the orientation of the northern basin border then this border must have a jog. Some people (e.g. Burkhard) explained this jog as a NS-striking strike-slip fault. But in reality we do not have any data on this. The seismic data of NRP20 are inconclusive on this.

It is, of course, true that a continuous basin is not proven, but most geological studies assume, or propose, a continuous basin, or depositional environment, between the Morcles and Doldenhorn regions. Also Burkhard, 1988, in his figure 3a, proposed a continuous deposition of the Malm limestone between the Morcles and Doldenhorn region. For simplicity of our numerical model, we assume that the basin is linear and linearly shallowing towards the Doldenhorn region. But of course, in reality this basin could have had a much more complicated internal geometry that was potentially responsible for the miss-alignment of the hinge lines.

154 The statement "absence of significant nappes in the Infrahelvetic complex" does not correspond to reality: there are three major nappes (Kaminspitz, Calanda and Tschep), all of which have significant displacements. But what they lack are recumbent folds, a fact that reflects the absence of an Early Jurassic basin and the Middle Jurassic sediments being very thin in comparision to the Late Jurassic and Cretaceous carbonate sequences.

We agree and we actually wanted to state what the reviewer says. We, hence, replaced 'which explains the absence of significant nappes in the Infrahelvetic complex below the Glarus thrust.' with 'which explains the absence of significant recumbent fold nappes in the Infrahelvetic complex below the Glarus thrust.'

155 The Doldenhorn nappe is much much closer in style to the Morcles nappe; it displayes long inverted limbs which are absent in the Glarus nappe complex in eastern

Switzerland.

We agree and we reformulated the sentence.

160-162 This interpretation is contested sharply by Pfiffner et al. (2011).

We are aware that there are different, partly opposing, interpretations. We clarify this. In our interpretation, the regions around Mt Joly and Aravis are part of the Subalpine chain. However, we follow the interpretation of Epard 1990 (and also Collet,1943) and argue that this region of the Subalpine chain has been deposited in the same North Helvetic basin as the Morcles nappe. Therefore, in our simplified model, the Subalpine chain results also from extrusion out of the North Helvetic basin during its closure.

509 The Chamonix-zone does not show a synclinal structure in nature. The Mesozoic sediments show a consistent younging from the Mont Blanc massif to the NW. The youngest sediments then but against the Aiguilles Rouges massif's basement. This is clearly visible form the structural maps 1:100'000 that you cite earlier on (Pfiffner et al. 2010).

We are aware that Pfiffner et al. 2010 do not interpret the Chamonix-zone as a syncline. However, several geological maps clearly show a synclinal structure with respect to the younging direction of the cover sediments of the Aiguilles-Rouges and Mont Blanc massifs (e.g. Paréjas, 1922; Ayrton, 1980; Oulianoff, 1924; Ayrton et al., 1987). These younging directions, indicating a synclinal structure, have also been verified in the field by one of the authors. We, therefore, maintain our interpretation in the manuscript, but mention the different interpretation of Pfiffner et al., 2010.

498 This statement about the Helvetic nappes needs a reference.

We added a reference to Pfiffner 1993 and reformulated the sentence.

512/13 Fig 14 The intention of this figure is much appreciated. I have some worries though if strain ellipses determined from pressure shadows are directly compared to strain determined from deformed oolites. And what are the contributions to the figure

by Bastida, Dietrich & Casey, Casey & Dietrich? Strains or cross-sectional geometry? Or are all the strain data from Ramsay & Huber? And I miss the effect of the Permo-carboniferous graben in the Aiguilles Rouges massif (it is partially inverted and folds the Morcles thrust above.

We clarified the contributions of the different authors in the figure captions.

578-605 It is no surprise that the three cross-sections chosen from along strike give different results. Cross-section shown in Fig. 15a is from the Rawil depression where the Morcles-nappe is deeply buried in the subsurface and thus drawn by projection only. The top basement beneath is constrained by seismic data of NRP20 and thus explains somewhat the reduced thickness of the nappe. However, I never put a name to the basement uplifts because of the uncertainty involved and urge the authors to delete them. One could equally well put the names of the Gastern and Aar massif in their place. The cross-section is more reliable for the Wildhorn nappe since this nappe out-crops along the trace of the cross section. The cross section shown in Fig. 15c shows a completely different nappe – and I doubt very much that it should be called Morcles nappe. In fact the Morcles nappe in the type locality "Dent de Morcles" displays hinge lines of internal folds that climb westwards over the Aiguilles Rouges massif, crossing its crest line and then plunge towards the SW beneath the Chablais and the Chaînes Subalpines thrust sheet. The structures shown in Fig. 15c are merely in the same structural position relative to the Chaînes Subalpines thrust sheet. The uncertainty emanating from the construction of (balanced) cross-sections could be extracted from the numerous cross-sections drawn along the trace of the cross-section shown in Fig. 15b. One of the main reason for divergent solutions is the observation that the lower limb is more horizontal whereas the upper limb plunges with 30âŮę to the NE (see discussion in Pfiffner 1993, a reference referred to in the manuscript). There aren't many cross-sections constructed along curved hinge lines as is necessary in this situation. The one shown in Pfiffner (1993) is based on the construction of Langenberg et al. (1987) who did use curved hinge lines. The major effect of the differing plunges is the

thickness of the Morcles nappe. Curved hinge lines yield a thickness of ca. 5 km, the cross-section by Escher et al. (1993) used in Fig. 15B suggests 7 km.

As mentioned before, we clarified our interpretation concerning the Subalpine chain and the Morcles nappe. We agree that different projections can yield different thickness in cross sections. This is why we show different sections, also to raise awareness that thickness and geometries from cross section should not be considered as axiomatic data when comparing them with numerical models. We clarified the text.

609-613 Does not present a conclusion. For me one important conclusion is the next following sentence (Nappe detachment, transport . . ..)

We deleted line 610 - 612 to make the conclusion more concise.

618-619 The importance of fieldwork in such a scenario is common sense.

We modified the sentence from 'Consequently, the results emphasize the importance of geological field work and reconstructions of the initial geological situation before fold-and-thrust belt formation.' to "Consequently, the results emphasize the importance of the initial geological situation before fold-and-thrust belt formation."

623 This is the place that screams for a statement on the nature of the strain in and out of the cross-sectional planes.

We added information on the strain.

628-638 These are findings that should go into the abstract.

We modified this section, but keep it in the Conclusions. We think it is a main Conclusion of our study that our model, which is based on standard rheological models and ignores micro-scale processes such as grain size reduction but considers pre-Alpine extensional structures, can reproduce several first-order features of the Helvetic nappe system.

Technical corrections

18 French-Swiss Alps is not a common denomination. I suppose you wish to include the Haute Savoie part of France. I suggest "Central Alps of France and Switzerland"

74, 108 see 18

Ok, we preplaced French-Swiss Alps by Central Alps of France and Switzerland

76-77 "laterally" instead of "along the lateral direction"?

Ok, we re-worded the sentence accordingly.

111 I suggest "Glarus nappe complex of eastern Switzerland"

Ok, we re-worded the sentence accordingly.

148 Diablerets

Ok, we re-worded the sentence accordingly.

326 2D numerical cross sections: why specify "numerical" here? All is numerical. And wouldn't "thrust sheet" be a better term than "thrust nappe", particularly as it is opposed to "fold nappe"?

We just want to be clear, otherwise for some readers it might not be clear whether we talk about geological or numerical cross sections. We just want to use the same term "nappe", since otherwise some readers might ask what is the difference between a nappe and a sheet.

496-497 suggestion: The resulting model nappe stack shows laterally major structural differences.

Ok, we re-worded the sentence accordingly.

515 It would be better to formulate what is observed, and not what is not observed (contact to basement)

We also would like to mention the differences between model and geological sections.

548 report (not reports)

Ok

549 frontal part

Ok

551 Doldenhorn and Glarus nappes (or Doldenhorn nappe and Glarus nappe complex)

Ok, we re-worded the sentence accordingly.

554 suggest

Ok, we re-worded the sentence accordingly.

563 start a new paragraph with "In terms of. . ..

Ok.

624 modeled by a stress cut-off at 40 MPa (instead of "due to")

Ok, we re-worded the sentence accordingly.

Please also note the supplement to this comment:
https://www.solid-earth-discuss.net/se-2019-173/se-2019-173-AC1-supplement.pdf

---

## Author Comment (AC2) · 4 Mar 2020

Author's response to
Control of 3D tectonic inheritance on fold-and-thrust belts: insights from 3D numerical models and application to the Helvetic nappe system by Spitz et al. Submitted for publication to Solid Earth

General comments The paper uses 3D numerical thermo-mechanical modelling of a heterogenous rifted margin with variable basement/sediments and structures assuming viscous-plastic laws. Results are then compared to deformation observations along 3 sections of the external Swiss-French Alps that have reached temperature conditions of 250-380âŮęC. The model setup is intended to reproduce the initial architecture of grabens of the European margin (proximal part) with different sediment thickness and geometry of basins varying along-strike. A V-shaped North-Helvetic basin is assumed. One main implications of such modelling approach is that for sedimentary units to be detached above the basement and form thrust nappes (little internal deformation) no mechanical softening is required. Strain localization results from the geometry and strength vari-ations, which conditions are likely met in many mountain belts. The over-all modelling presentation and results are well written and quite easy to follow. I am only concern about erosion that is not modelled (see below). Results are sounding and the application to the Alps relevant, but please consider say some words about the choice of having discarded the role of erosion. I only suggest to consider reorganising/rewriting the Introduction and the Discussion (Section 5.3).

Specific comments Introduction Lines#44-50: This paragraph is very specific (i.e. only viscous mechanism are addressed) relative to the rest of the introduction. I suggest to move them after lines#55-61 where the authors present older studies with more gen-eral mechanical behavior.

We changed the order as suggested.

In addition, in classical model of FTBs (like later in the intro) a major decoupling level (low friction or linearly viscous like salt) if often assumed. The high contrast between basal and internal strength allows thrust nappe formation. This best applies to external zones when pressure and temperature (well below 300âŮęC) does not modify the original rheological layering. In the case of shear zones the au-thors study here (intermediate domains of tectonic burial) this is very different because such weak layers where deformation localizes is not prescribed so a "self-localization" process is required. The introduction should better emphasize the differences between the end-member approaches. One "cold" frictional classical approach of Coulomb-type thrust wedges vs one "hot" viscous-plastic approach of ductile nappe stacking. They are both valid and should co-exist, depending which part of the orogen you are dealing with. Here we are clearly more interested in ductile-type fold-thrust belts. Methods. The authors should indicate why erosion is not relevant or not taken into account in their modelling approach. Because erosion is not considered here rocks do not cool during deformation, thrusting/folding and the crust is thickening. Therefore they are always in the ductile field. This may be valid but should be clearly presented especially the authors are dealing with the most recent Miocene sequence of shortening in Alps and erosion is a major factor in orogens during these late stages.

There are three reasons why we do not model erosion: (1) due to simplicity. If we would consider erosion, we should ideally test the impact of different erosion models (diffusion-type, slope dependent etc.) on our results, which is beyond the scope of the study. (2) the topography in our model does not show significant lateral variation and with the exceptions of minor undulations the topography remains more or less straight. (3) the main phase of uplift and associated exhumation started at ca. 20 Ma, which is towards the end of the main phase of nappe formation. Therefore, erosion, exhumation and associated cooling might have only affected the late stage of nappe formation. We added some of this information in the revised text.

Discussion The authors are able to reproduce thrust nappes (little internal deformation) of sedimentary units detached above the basement without mechanical softening. This is important. "Kinematic" strain localization is the result of the geometry and internal strength variations.

Lines#453-456: Section 5.1. This part raises a very interesting point. The impact of this result is that the initial rheological layering and configuration of these layers suffice to produce thrust nappes. I am wondering how this could extend to the application of "static" models of brittle thrust wedges (e.g. Dahlen et al., 1990) with no mechanical softening or more dynamic ones including friction (Ruh et al., 2012). These models indeed assume that mechanical properties do not change with time and incremental displacement.

Indeed, the results of Ruh et al. (2012) and ours (and several other studies) essentially show that material heterogeneities can generate localized deformation and when such heterogeneities are considered, it is not necessary to apply prominent strain softening to the model units. In our case, we argue that the particular geometry of the nappe system, and in particular it's lateral variation, are to first order determined by the pre-Alpine geometry, or tectonic inheritance. This seems maybe obvious and has been suggested by many geologists, however, no one has, to the best of our knowledge, supported this with 3D thermo-mechanical numerical simulations.

Lines#463-470: Thickness does matter to explain salients and recesses in FTBs but these peculiar structure may also reflect the occurrence of laterally discon-tinuous decollement levels (changes in lithology and thickness - for viscous décollement - also play an important role); in other words not only changes in the overburden thickness are relevant.

We clarified the statement and now write: 'This observation is in accordance with our study which shows that lateral changes in the lithology, such as thickness and rheology, produce different salients (\ref{fig:fig4}).'

The Section 5.3 presents a discussion on our attempts to compare geological sections with numerical models. I had hard time to follow the reasoning here. First the authors introduce the balanced cross-section approach which can only scarcely be applied here because of the dominant ductile behavior of the HN; thickening/thinning of layers (and lengths variations) during deformation and out-of-plane deformation preclude the use of 2D balancing techniques. 3D balancing techniques with volume balancing could do it. I think the value of this section is not the comparison with balanced cross-sections. I would suggest to focus on the 3D aspects of their numerical approach and explain how the structural complexities seen in 3D modelling could be simply explained by lateral variations in the original structure rather than by mechanical complexities mechanics. This is important when searching for mechanical interpretation of lateral changes in cross-sections (balanced or not !). Maybe is what the authors intended to do here but could not clearly see it from my reading.

We reformulated this section to make our aims clearer.

Other technical corrections are in the pdf attached.

Please also note the supplement to this comment:

https://www.solid-earth-discuss.net/se-2019-173/se-2019-173-RC2-supplement.pdf

We considered all corrections and modified the manuscript accordingly.

Please also note the supplement to this comment:
https://www.solid-earth-discuss.net/se-2019-173/se-2019-173-AC2-supplement.pdf

---

## Editor Comment (EC1) · Susanne Buiter (Editor) · 20 Mar 2020

Dear authors,

Many thanks for the replies to the reviews and the revision of your manuscript. I acknowledge the challenges of 3D numerical models, that include plasticity, and the direct application of insights learned to natural systems, so please take the following as constructive input from my side for your manuscript. As you will know (since one of the co-authors is involved), Solid Earth recently published a manuscript "Tectonic inheritance controls nappe detachment, transport and stacking in the Helvetic nappe system, Switzerland: insights from thermomechanical simulations" by Kiss, Duretz and Schmal-

holz (https://www.solid-earth.net/11/287/2020/). This paper considers the same area and also uses numerical experiments to gain insight in the Helvetic nappe system, but in 2D. I noticed that the setup is very similar to the one in your manuscript. What i would like to ask is the following:

(1) To discuss the similarities and differences between the study of Kiss et al and your study. It would allow an elaboration of what we learn from using 2D versus 3D experiments in a complex area.

(2) A verification or statement that a 2D version of your model would give similar results to Kiss et al (2020), or if not, why not.

(3) and more minor, could you discuss whether the properties and thickness of the sticky air layer follow the recommendations by Crameri et al (2012, GJI 189)?

Many thanks, Susanne Buiter

---

## Referee Report (RR1)

Comments to
Spitz et al        se-2019-173-manuscript-version3

**Control of 3D tectonic inheritance on fold-and-thrust belts: insights from 3D numerical models and application to the Helvetic nappe system**

The authors have taken the suggestions into account and improved the manuscript significantly.
I have two aspects that the authors should consider:

1) The reference Ayrton 1980 is incomplete. The article appeared in Eclogae geological Helvetiae, vol. 73/1, 137-172

This article is about the Chamonix zone, which fwas formerly called Chamonix syncline. Since this point is one disagreement between the authors and myself I took the time to read Ayrton 1980 once again very carefully. It turns out as I said earlier on that most of the Triassic-Jurassic-Cretaceous cover of this zone pertains to  the Mont Blanc massif and is subvertical to overturned. There are some Jurassic sediments adjacent to the Aiguilles Rouges massif. Between the two there is a socalled „median zone" which would today be called a tectonic mélange; locally it is mylonitic.
So much for a „syncline". The authors might have second thoughts on what they publish.

2) The publication Pfiffner et al. 2011 actually consists of two things:
a) The map sheets, authors are Pfiffner et al. as cited, but the year is 2010
b) The Explanatory Notes, written by Pfiffner and published in 2011.
The authors might wish to differentiate in order to give a correct reference. To help I list the lines where reference to the maps is correct and where the Explanatory notes give a better reference:
Pfiffner et al 2010: lines 91, 93, 161, 523
Pfiffner 211: lines 66, 70, 170, 538

Apart from this the manuscript can be published.

Adrian Pfiffner

---

## Author Response (AR2)

Spitz et al se-2019-173-manuscript-version3

Control of 3D tectonic inheritance on fold-and-thrust belts: insights from 3D numerical models and application to the Helvetic nappe system
The authors have taken the suggestions into account and improved the manuscript significantly.
I have two aspects that the authors should consider:

1) The reference Ayrton 1980 is incomplete. The article appeared in Eclogae geological Helvetiae, vol. 73/1, 137-172

We added the correction for the reference Ayrton 1980.

This article is about the Chamonix zone, which was formerly called Chamonix syncline. Since this point is one disagreement between the authors and myself I took the time to read Ayrton 1980 once again very carefully. It turns out as I said earlier on that most of the Triassic-Jurassic-Cretaceous cover of this zone pertains to the Mont Blanc massif and is subvertical to overturned. There are some Jurassic sediments adjacent to the Aiguilles Rouges massif. Between the two there is a socalled „median zone" which would today be called a tectonic mélange; locally it is mylonitic. So much for a „syncline". The authors might have second thoughts on what they publish.

We re-formulated the paragraph concerning the Chamonix zone. In fact, we want to argue that the structural observations around the Chamonix zone are in agreement with the modelled ductile closure of the half-graben between the Mont Blanc and Aiguilles Rouges massif. Especially the subvertical to overturned Mont Blanc cover supports the internal deformation of the Mont Blanc massif. The sedimentary cover of the Aiguilles Rouges shows locally also a subvertical orientation with a younging direction opposite to the one of the Mont Blanc cover. Therefore, the term syncline is sometimes used because of the opposite younging direction. However, we removed the term syncline now since it may be misleading with respect to the kinematic evolution. The modified paragraph is now:

"The strong layer is still connected to the adjacent basement horst and the deformed internal weak units, highlighted by the green passive marker lines, resemble a recumbent fold. The inner part of the fold nappe roots into a steep cusp, which is analogues to the Chamonix zone between the Aiguilles Rouges and Mont Blanc massif (Figure 13a). The Chamonix zone is characterized by subvertical to overturned sediments representing the cover of the Mont Blanc massif and locally by subvertical sediments representing the cover of the Aiguilles Rouges massif (Paréjas, 1922; Oulianoff, 1924; Ayrton, 1980; Epard, 1986; Ayrton et al., 1987; Pfiffner et al., 2011). The overturned cover sediments of the Mont Blanc massif suggest that the Mont Blanc massif did not behave as an effectively rigid basement block but was internally deformed (e.g., Epard, 1986). The subvertical cover sediments of both massifs, having opposite younging directions, support our numerical model, which shows an overall ductile closure of the half-graben. Consequently, we argue that the half-graben was not closed by dominantly brittle inversion tectonics during which the Mont Blanc massif was thrust as an effectively rigid block above an effectively rigid Aiguilles Rouges massif (e.g., Granado and Ruh, 2019)."

2) The publication Pfiffner et al. 2011 actually consists of two things:

a) The map sheets, authors are Pfiffner et al. as cited, but the year is 2010
b) The Explanatory Notes, written by Pfiffner and published in 2011.

The authors might wish to differentiate in order to give a correct reference. To help I list the lines where reference to the maps is correct and where the Explanatory notes give a better reference:
Pfiffner et al 2010: lines 91, 93, 161, 523
Pfiffner 211: lines 66, 70, 170, 538

We added/corrected the suggested citations for the publication of Piffner et al. 2011 and Piffner 2011.
However, we cite the map sheets still with the year 2011, because google schoolar and also the BORIS (https://boris.unibe.ch/10644/) provide the year 2011.

Apart from this the manuscript can be published.

Adrian Pfiffner

Dear authors, Many thanks for the replies to the reviews and the revision of your manuscript. I acknowledge the challenges of 3D numerical models, that include plasticity, and the direct application of insights learned to natural systems, so please take the following as constructive input from my side for your manuscript.

As you will know (since one of the co-authors is involved), Solid Earth recently published a manuscript "Tectonic inheritance controls nappe detachment, transport and stacking in the Helvetic nappe system, Switzerland: insights from thermomechanical simulations" by Kiss, Duretz and Schmalholz (https://www.solid-earth.net/11/287/2020/). This paper considers the same area and also uses numerical experiments to gain insight in the Helvetic nappe system, but in 2D. I noticed that the setup is very similar to the one in your manuscript.

What i would like to ask is the following:

(1) To discuss the similarities and differences between the study of Kiss et al. and your study. It would allow an elaboration of what we learn from using 2D versus 3D experiments in a complex area.

We added several sentences in the discussion.

(2) A verification or statement that a 2D version of your model would give similar results to Kiss et al (2020), or if not, why not.

We added several sentences in the discussion.

(3) and more minor, could you discuss whether the properties and thickness of the sticky air layer follow the recommendations by Crameri et al (2012, GJI 189)?

We added a statement in the model configuration section that our sticky air parameters follow the recommendations of Crameri et al..

[revised manuscript text omitted]